# Dose optimization of TBI-223 for enhanced therapeutic benefit compared to linezolid in antituberculosis regimen

Natasha Strydom [1], Jacqueline P. Ernest[1], Marjorie Imperial[1], Belén P. Solans [1], Qianwen Wang[1], Rokeya Tasneen[2], Sandeep Tyagi[2], Heena Soni[2], Andrew Garcia[2], Kristina Bigelow[2], Martin Gengenbacher [3,4], Matthew Zimmerman[3], Min Xie [3], Jansy P. Sarathy [3], Tian J. Yang [5], Véronique Dartois [3,4], Eric L. Nuermberger [2] & Radojka M. Savic [1] ✉

TBI-223, a novel oxazolidinone for tuberculosis, is designed to provide improved efficacy and safety compared to linezolid in combination with bedaquiline and pretomanid (BPaL). We aim to optimize the dosing of TBI-223 within the BPaL regimen for enhanced therapeutic outcomes. TBI-223 is investigated in preclinical monotherapy, multidrug therapy, and lesion penetration experiments to describe its efficacy and safety versus linezolid. A translational platform incorporating linezolid and BPaL data from preclinical experiments and 4 clinical trials (NCT00396084, NCT02333799, NCT03086486, NCT00816426) is developed, enabling validation of the framework. TBI-223 preclinical and Phase 1 data (NCT03758612) are applied to the translational framework to predict clinical outcomes and optimize TBI-223 dosing in combination with bedaquiline and pretomanid. Results indicate that daily doses of 1200–2400 mg TBI-223 may achieve efficacy comparable to the BPaL regimen, with >90% of patients predicted to reach culture conversion by two months.

Tuberculosis treatment requires new therapies that offer shorter duration, safer regimens, and more potent drug combinations to decrease the incidence of tuberculosis[1]. The oxazolidinone class of antibiotics has shown great value in the bedaquiline, pretomanid, linezolid (BPaL) regimen in recent Phase 3 Nix-TB and ZeNix trials (NCT03086486)[2]. Nix-TB enrolled 109 patients with multidrug-resistant tuberculosis who received a combination of bedaquiline at a dose of 400 mg once daily (QD) for 2 weeks followed by 200 mg three times a week for 24 weeks, pretomanid at a dose of 200 mg daily for 26 weeks, and linezolid at a dose of 1200 mg daily for up to 26 weeks, with allowable dose adjustments depending on toxicity. After 6 months of treatment, 90% of patients had favorable outcomes[2].

However, linezolid has safety concerns and was originally not intended for long-term use (greater than 28 days) due to its associated inhibition of mitochondrial protein synthesis leading to bone marrow myelosuppression and neuropathic effects[3]. Linezolid dose reduction was necessary for most patients in the Nix-TB trial due to safety events with 81% of patients experiencing mild-to-moderate peripheral neuropathy and 48% experiencing myelosuppression[2]. The ZeNix trial tested lower linezolid doses and variations in treatment duration. A lower 600 mg linezolid dose limited to a 9-week treatment duration had the best safety with 7% myelosuppression and 13% peripheral neuropathy compared to 22% myelosuppression and 38% peripheral neuropathy for the 1200 mg linezolid given for 26 weeks trial arm. However, this safer

[1]Department of Bioengineering and Therapeutic Sciences, Schools of Pharmacy, University of California, San Francisco, CA, USA. [2]Center for Tuberculosis Research, Division of Infectious Diseases, Department of Medicine, Johns Hopkins University, Baltimore, MD, USA. [3]Center for Discovery and Innovation, Hackensack Meridian Health, Nutley, NJ, USA. [4]Hackensack Meridian School of Medicine, Hackensack Meridian Health, Nutley, NJ, USA. [5]TB Alliance, New York, NY, USA. ✉e-mail: rada.savic@ucsf.edu

dose and duration had an efficacy tradeoff of 84% compared to 93% favorable outcomes for the higher dose and longer duration[4].

Linezolid has a narrow therapeutic window, as evidenced by the concentration required for effective *Mycobacterium tuberculosis* eradication and the levels associated with patient toxicity. A pharmacokinetic-toxicodynamic model, that utilized population pharmacokinetic modeling to simulate dynamic individual linezolid concentrations linked to toxicity models, identified a peripheral neuropathy $IC_{50}$ of 1.3 mg/L for linezolid[5]. Median plasma concentrations from patients in the Nix-TB trial exceeded this threshold across all dosages, from 300 mg twice daily to 1200 mg QD, with a minimum concentration of 2.1 mg/L at the lowest dose[5]. These findings based on clinical data and modeling and simulations are consistent with previous studies that showed mitochondrial toxicity and safety events were associated with trough levels above 2 mg/L[6]. Furthermore, the minimum inhibitory concentration (MIC) of linezolid necessary to inhibit 90% of *Mycobacterium tuberculosis* isolates range from 0.25 mg/L to 1.0 mg/L[7–10], underscoring the limited range within which linezolid can be both effective and safe.

The mechanism of action of oxazolidinones is to target bacterial ribosomes and inhibit bacterial protein synthesis[11]. Mitochondrial protein synthesis in humans shares a conserved ribosomal structure and oxazolidinones generally have poor selectivity between bacterial and mitochondrial ribosomes leading to disruption of mitochondrial protein synthesis and bone marrow myelosuppression. Oxazolidinones that have the highest antibacterial potency also show the greatest inhibition between mitochondria and prokaryotes, implying the targeted ribosomal binding sites between eukaryotes and prokaryotes are very similar[12]. Thus, designing novel oxazolidinones with similar or improved potency and selectivity toward bacterial ribosomes is challenging.

A safer, more selective oxazolidinone candidate that can provide similar efficacy to linezolid is therefore needed to reduce the likelihood of adverse events. TBI-223 is a potential candidate for replacing linezolid in the BPaL regimen which, despite its lower in vitro activity, has shown an improved toxicity profile in preclinical toxicology studies and potentially improved therapeutic window[13,14].

We, therefore, aimed to investigate the performance of TBI-223 in preclinical monotherapy, multidrug therapy, and lesion penetration experiments and describe its efficacy profile as compared to linezolid (Fig. 1). We used a data-driven modeling approach that integrated preclinical and early clinical data to further optimize TBI-223 dosing amount, frequency and formulation to provide more effective TB regimens. This translational platform was built using linezolid data from preclinical experiments and 4 clinical trials that allowed us to validate the framework and confirm that predictions were in line with previous linezolid clinical outcomes and clinical trial results. Using this integrated framework, we propose a TBI-223 dose that provides equal or even improved effectiveness to the linezolid 1200 mg dose as used in Nix-TB study, or 600 mg dose used in the ZeNix study. Additionally, we believe new and existing drug regimens should be systematically evaluated for lesion distribution as achieving adequate exposures at the site of infection is crucial for successful treatment[15].

## Results

### Compiled data for monotherapy linezolid and TBI-223
Data were generated and collated from multiple animal experiments (Fig. 2). Preclinical pharmacokinetic (PK) data were collected from mice (BALB/c, and C57BL/6), Sprague-Dawley rats, and beagle dogs. These data were originally used to provide dose recommendations for the TBI-223 Phase 1 human trial, (NCT03758612), and details are contained in Fig S1. Monotherapy pharmacokinetic-pharmacodynamic (PK-PD) data from BALB/c mice and included linezolid and TBI-223 experiments performed at the same time and under the same conditions with 12 doses tested for each drug ranging from 7.2 mg/kg to

335 mg/kg for linezolid and 21.4–1000 mg/kg for TBI-223[16]. Previous linezolid experimental data that utilized the same dose fractionation in a chronic infection model[16] were also collected and modeled with details in supporting Table S1. Combination PK-PD data of TBI-223 and linezolid with bedaquiline 25 mg/kg QD and pretomanid 25 mg/kg twice daily (BID) were generated in BALB/c mice in a 17-day infection model. Four doses were tested for each oxazolidinone and ranged from 12.5 to 50 mg/kg BID and 100 mg/kg QD for linezolid and 15–45 mg/kg BID and 100 mg/kg QD for TBI-223.

Lesion penetration experiments were performed in New Zealand White rabbits and sampled three major tissue types including uninvolved lung, and cellular and caseous lesions. Total drug concentrations were determined by LC-MS/MS using homogenized samples of these tissue types or samples obtained by laser capture microdissection to determine the spatial concentration in the inner and outer regions of cellular and caseous lesions.

In total, 1308 data points were generated across preclinical experiments, across 21 dose experiments. The final results and model parameters of interest are shown in Tables 1 and 2. Additional PK-PD data in longer incubation mouse models were available for linezolid and these model parameters are reported in Table S1. Structural model diagrams and visual predictive checks of the models are shown in Figs S1 and S3–S6, respectively, and show that the models captured the available preclinical data.

### Comparison of linezolid prediction with clinical endpoints
Simulations across monotherapy and combination therapy, and lesion distribution are included in Fig. 3. Simulations of CFU over time of linezolid 600 mg administered QD or BID compared well to previously published values with a slight over-prediction of efficacy beyond the first 2 days, Fig. 3A[17]. For combination therapy simulations individual patient-level data were used to provide accurate baseline bacterial burden, individual PK parameters, and sampling times. The model performed well at predicting bactericidal activity as determined by CFU measurements in combination with bedaquiline and pretomanid in the first 28 days of treatment in the Nix-TB trial, Fig. 3B, with majority of the observed data falling within the 95% prediction intervals, and the median of the observed data aligning closely with the prediction median of the model.

In Fig. 3C, simulations of sputum culture conversion are depicted alongside observed Nix-TB trial data in a Kaplan–Meier curve. The model accurately reflected the progression of culture conversion, matching the observed probabilities of positive status with those predicted at individual time points. The dosing history of the Nix-TB trial and its dose interruptions and adjustments are illustrated in Fig. S7A. Modeling the concentration response of linezolid in combination with bedaquiline and pretomanid revealed a narrow therapeutic window when compared to linezolid toxicity parameters from a published red blood cell toxicity model[5], Fig. S7B. At steady state, the 600 mg QD dose achieves the linezolid concentration needed for 95% of its maximal effect but also activates this toxicity pathway during the dosing interval.

For ZeNix trial predictions, patient-level pharmacokinetics and baseline TTP were used, and sampling times adjusted to that of the trial protocol. Similar to the Nix-TB trial data, the model could predict probability of culture conversion well, Fig. 3D.

To compare with data from the NCT00816426 trial that examined lesion concentration in lung specimens obtained from patients with tuberculosis undergoing lung resection surgery, we simulated a matching daily dose of 300 mg linezolid with patient population variance and lesion PK parameters from the rabbit model and found that the trial observations were within range of the simulations (Fig. 3E).

### Simulations to optimize TBI-223 dose, frequency, and formulation
To provide the best conditions for TBI-223 efficacy, we evaluated different formulations, dose frequencies, and dose amounts. TBI-223

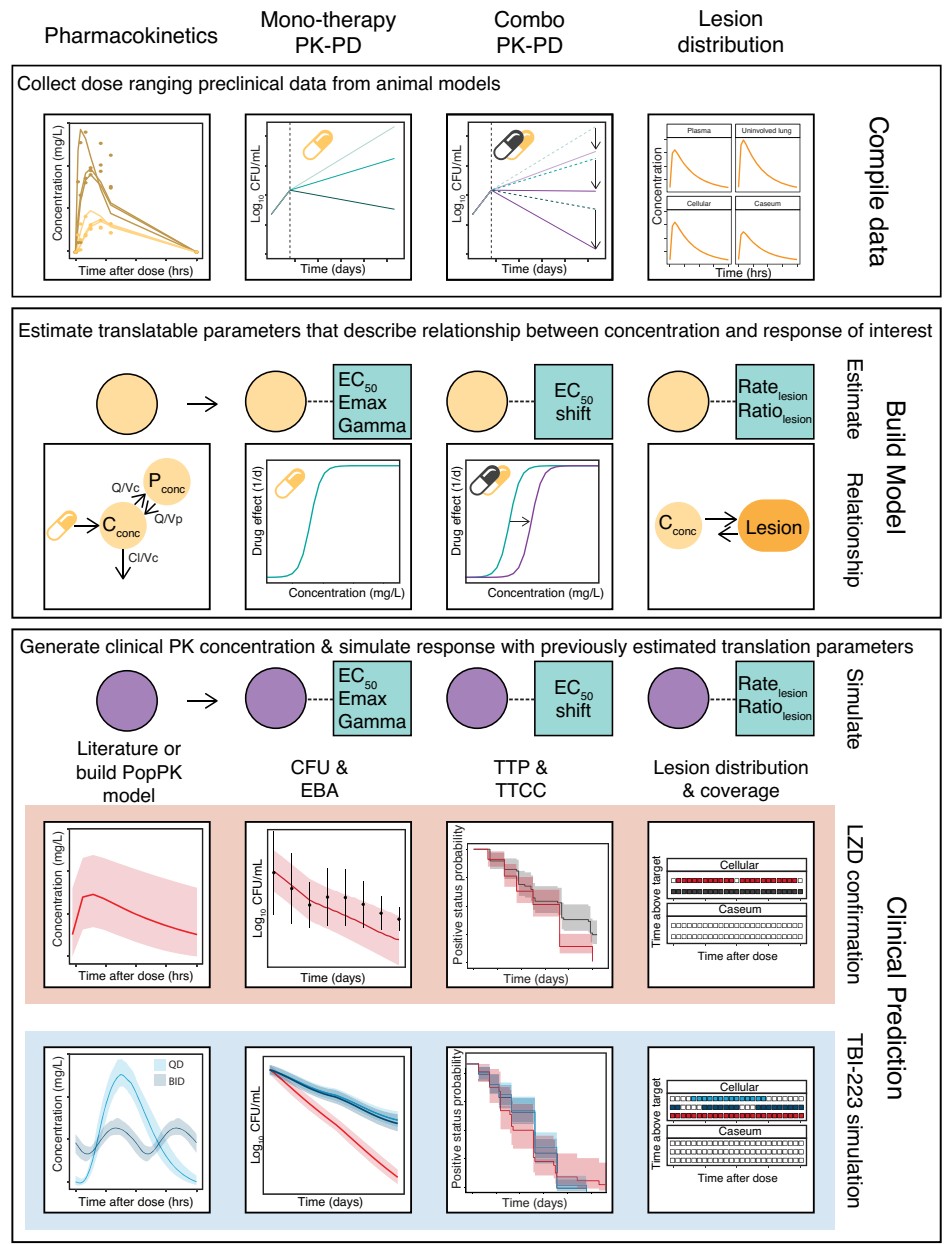

**Fig. 1 | Project scope and workflow.** The modeling workflow consisted of three major parts: (i) compiling data and tools from preclinical and clinical studies, (ii) building a model to relate concentration to the response of interest (e.g., efficacy or lesion distribution), and finally, (iii) substituting preclinical pharmacokinetic (PK) concentrations in our framework with clinical population PK concentrations to predict clinical response. This was done for linezolid (red) and TBI-223 (blue). Linezolid data was used to confirm that each respective model showed adequate translation of clinical outcomes. Linezolid was also used in comparison to TBI-223 to assess whether TBI-223 can provide similar efficacy to linezolid. LZD linezolid, PK-PD pharmacokinetic-pharmacodynamic, CFU colony-forming units, Q inter-compartmental clearance, Vc the central volume of distribution, Vp the peripheral volume of distribution, $C_{conc}$ central compartment concentration, $P_{conc}$ peripheral compartment concentration, $EC_{50}$ half-maximal effective concentration, Emax maximum efficacy, TTP time to positivity, TTCC time to culture conversion.

modeling used population PK-based data from a Phase 1 trial in healthy volunteers that examined several dosing formulations including immediate-release (IR) and sustained-release (SR) capsules, Fig S1. Simulations were limited to previously prioritized IR1 and SR1 formulations, hereafter simply referred to as IR and SR respectively. A comparison of IR and SR efficacy in combination therapy is shown in Fig. 4A and B. The SR formulation was predicted to have slightly higher efficacy compared to IR. However, a total dose above 2400 mg provides a marginal improvement in efficacy due to the dose-response of TBI-223 being near its maximum level of efficacy. A co-formulated dose of IR 600 mg and SR 1800 mg provides the best overall predicted efficacy for a 2400 mg dose (total decrease in CFU over 7 days was equal to 0.658) (Fig. 4C). However, this improvement is small (less than 2% overall increase) compared to an SR 2400 mg dose resulting in a 7-day decrease in CFU of 0.647. Improved lesion coverage in cellular lesions, defined as the proportion of the dosing interval for which local concentrations exceed the TBI-223 $IC_{90}$ identified in an in vitro macrophage infection model, was seen when using a co-formulation approach. For example, administering IR 900 mg and SR 1500 mg together had 5% improved coverage compared to an SR 2400 mg dose (Fig. 4D). At a total dose of 2400 mg, splitting the dose for twice daily administration improves the decrease in CFU by 4.5% compared to once-daily dosing and provides improved coverage in cellular lesions. At lower doses, this difference

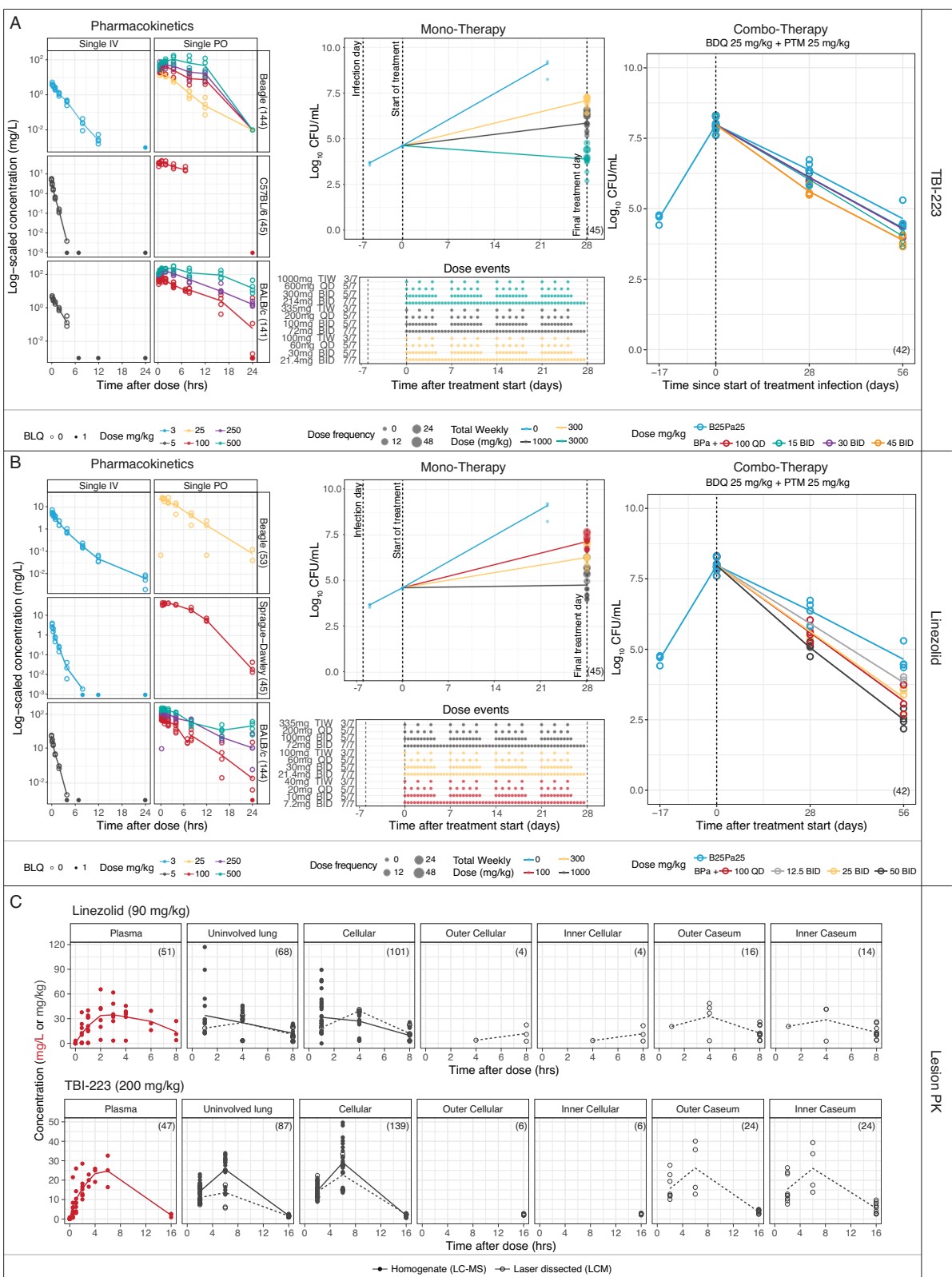

**Fig. 2 | Compiled preclinical data for TBI-223 and linezolid.** All collected preclinical data showing drug concentrations, dose events, and CFU counts, including data below the limit of quantification for TBI-223 (**A**) and linezolid (**B**). Lesion drug concentration data collected from rabbits are shown in (**C**). For all panels, individual observations are shown as dots with mean values shown as lines. Starting on the left of (**A**, **B**), preclinical pharmacokinetic data were collected to describe pharmacokinetics in a multispecies model and predict first-in-human doses. In the middle sections of these two panels, the monotherapy efficacy experiments are displayed with CFU counts on the top and their equivalent dosing events below to illustrate the dose fractionations that were used. Both the TBI-223 and linezolid plot display the same control as a reference point. Similarly, on the right, the combination efficacy experiment is shown with the control experiment shown in both plots for reference. For the lesion experiments in panel C, the plasma data points are shown in red and each lesion compartment is shown as a panel with quantification methods identified by shape and linetype. The number of available data points per compartment is shown in parentheses. PK pharmacokinetics, CFU colony-forming units, BDQ bedaquiline, PTM pretomanid. Source data are provided as a Source Data file.

was larger, up to 13% improvement when comparing a 600 mg BID to a 1200 mg QD dose.

## Dose recommendations for TBI-223 and its comparison to linezolid

TBI-223 efficacy endpoints as compared to linezolid are shown in Fig. 5. The monotherapy concentration-response profiles of linezolid and TBI-223 reveal that linezolid had a significantly higher $E_{max}$ than TBI-223. Simulated 2-day CFU response for TBI-223 at a dose 4 times as high as linezolid was predicted to cause less than half the reduction in CFU as compared to linezolid 600 mg daily (Fig. 5A). In combination, however, the difference in $E_{max}$ is reduced between linezolid and TBI-223. The reductions in CFU over time for TBI-223 dosed at 2400 mg and linezolid 600 mg QD in combination with bedaquiline and pretomanid were, therefore, closer (monotherapy had a CFU difference of 0.25 between LZD and TBI-223 while combination therapy showed a difference of 0.09 between LZD and TBI-223) (Fig. 5B). Twice daily dosing of 900 mg or more of TBI-223 had the same level of efficacy when compared to QD 2400 mg as TBI-223 reached its maximum efficacy level. The dose-response simulations of TBI-223 (Fig. 5C) showed that TBI-223 cannot match linezolid efficacy at any dose if linezolid is given without interruption. Increasing the dose of TBI-223 above 2400 mg had a negligible improvement in efficacy. The difference in CFU decrease of linezolid in combination with bedaquiline and pretomanid is relatively small between 600 mg and 1200 mg QD and the higher dose did not significantly improve outcomes for the median population. However, higher doses did minimize population variability as PK levels are above $EC_{50}$ for a larger proportion of the population (Fig. 5A).

To evaluate the impact of TBI-223 in combination with bedaquiline and pretomanid on culture conversion, we simulated time-to-culture conversion and compared it to the ZeNix arms. A Kaplan–Meier curve of TBI-223 compared to clinical data of the ZeNix trial showed that TBI-223 could provide similar culture conversion in patients during treatment (Fig. 5D). These simulations show that if uninterrupted, TBI-223 in combination with bedaquiline and pretomanid can outperform linezolid. These simulations predict that >90% of patients will experience culture conversion by two months.

The final metric in the efficacy profile of TBI-223 was lesion distribution and coverage. Examining lesion penetration coefficients for TBI-223 showed values that were similar to the estimates for linezolid (Table 2). Similarly, the in vitro inhibition targets for TBI-223 and linezolid were similar, with relatively low macrophage $IC_{90}$ values that were similar in range relative to MIC and little-to-no bactericidal activity against *M. tuberculosis* in the ex vivo caseum MBC assay for both drugs[18]. However, the overall lower concentrations of TBI-223 dosed at 2400 mg QD throughout the dosing interval meant that TBI-223 did not have the same 100% coverage as seen with linezolid dosed at 600 mg QD in uninvolved lung and cellular lesions. TBI-223 concentrations exceeded the respective targets for cellular lesion compartments for 50% of the dosing interval for 2400 mg QD while 1200 mg twice daily improved cellular lesion coverage to 75% (Fig. 5E). Full lesion profiles are presented in Fig S8. Neither linezolid nor TBI-223 concentrations exceeded the monotherapy caseum MBC in the caseum of lesions. However, the caseum MBC of TBI-223 in combination with equimolar amounts of bedaquiline and pretomanid shifted from 46.8 mg/L to 3.9 mg/L which TBI-223 concentrations reached in caseum lesions (Fig S8).

## Discussion

Providing the accurate assessment and translation of comprehensive preclinical and clinical data for novel drug candidates is an essential step before advancing novel compounds into clinical efficacy testing. A fully integrative approach can generate a meaningful rationale to justify the progression of a compound and generate useful data to

inform Phase 2 and 3 trials. The research strategy to provide this integrative approach was structured around a systematic selection of animal experiments, each chosen for its ability to yield data critical for predicting clinical outcomes. This included understanding TBI-223's efficacy in monotherapy and combination therapies and its pharmacodynamics at the infection site. The model was therefore an integration of data from these key animal studies, forming a composite framework that utilized in vivo data for clinical activity prediction and early clinical data, Fig. 1.

Using this data-driven modeling platform we predict that an SR formulation of TBI-223, administered QD at a dose of 2400 mg in combination with bedaquiline and pretomanid will provide adequate efficacy in early Phase 2 studies. Evaluating dose ranges from 1200 mg to 2400 mg should also generate data necessary to describe the dose-response profile of TBI-223 for subsequent clinical trials. While monotherapy TBI-223 is not as potent as linezolid in vitro and preclinical in vivo testing, when dosed in combination with bedaquiline and pretomanid, the gap in efficacy improved in favor of TBI-223.

Our results suggest the SR formulation should be selected based on improved lesion coverage and long-term efficacy. However, the IR formulation might provide better short-term early bactericidal activity (EBA) results as the high initial concentration provided a slight improvement in CFU reduction during the first two days of administration. Overall, the EBA and lesion coverage for TBI-223 could benefit from a co-formulation of IR and SR. However, the predicted clinical impact is not significant and the SR formulation by itself provides similar final outcomes. Using 1200 mg BID did improve lesion coverage compared to 2400 mg QD and could be more beneficial to reach hard-to-treat sites of disease, including the caseum lining cavity walls and in closed caseous foci. The dose-response simulations for TBI-223 showed that twice daily dosing could provide the opportunity to lower the dose to 900 mg twice daily (total daily dose of 1800 mg) and still have similar efficacy to 2400 mg daily dosing.

A 2400 mg dose of TBI-223 is predicted to produce concentrations above its respective $EC_{50}$ for a longer duration than linezolid dosed at 600 mg QD and allows TBI-223 to be dosed at its maximum efficacy. Linezolid cannot be dosed to have a higher minimum concentration value as this is highly associated with toxicity[5,6,19]. TBI-223 at 2400 mg QD still has a favorable safety margin of 16-fold based on TBI-223 exposures achieved in a 28-day rat toxicity study that showed no hematological changes or bone marrow toxicity at the highest tested dose (Table 2). Linezolid had a safety margin of only 0.69 fold due to observed bone marrow toxicity in the same animal model when compared to a dose of 600 mg QD which has similar daily exposure to TBI-223 2400 mg QD. The proposed doses demonstrated safety in short-term healthy volunteer studies and long-term safety will be determined in phase 2 trials.

While EBA for TBI-223 monotherapy is predicted to be low, it should not be the determining step for its advancement in further trials as the clinical predictions show that, in combination with bedaquiline and pretomanid, TBI-223 has much better potential. This shows that the attrition of drugs due to poor EBA in monotherapy could result in lost opportunities as having the right combination partners can greatly improve overall efficacy. Additionally, dose optimization should not be based on monotherapy as the dose-response for linezolid and TBI-223 changed significantly when combined with bedaquiline and pretomanid. Monotherapy dose escalation is necessary to assess the safety of new TB drugs. However, its usefulness in dose selection for what ultimately will be a combination therapy may be limited, as seen from our results.

For combination therapy estimation we tried several methods (including general pharmacodynamic interaction model (GPDI),[20] adapted and simplified GPDI approaches, additive bedaquiline, and pretomanid models) and found simply estimating effect parameters in the same fashion as our monotherapy models gave the best clinical

**Table 1 | Clinical model parameters**

| Parameter | LZD (%RSE) | TBI-223 (%RSE) |
|---|---|---|
| Clinical population PK parameters | | |
| Intrinsic clearance, Clint (L/h) | 7.9 (38)[a] | 32.8 (17.6) |
| Michaelis-Menten Constant, Km (mg/L) | 16 (90)[a] | 18.5 (20.3) |
| Central volume, Vc (L) | 49 (11)[a] | 110 (7.24) |
| Intercompartment clearance, Q (L/h) | 0.8 (90)[a] | – |
| Peripheral Volume, Vp (L) | 14 (40)[a] | – |
| Rate of absorption, ka (h$^{-1}$) | 1.1 (22)[a] | – |
| IR Mean transit time, Mtt (h) | – | 0.508 (40.5) |
| IR Rate of transit, Ktr (h$^{-1}$) | – | 3.51 (30.0) |
| IR Rate of absorption, ka (h$^{-1}$) | – | 2.24 (68.6) |
| SR Mean transit time, Mtt (h) | – | 5.32 (21.9) |
| SR Rate of transit, Ktr (h$^{-1}$) | – | 0.639 (29.6) |
| SR Rate of absorption, ka (h$^{-1}$) | – | 0.548 (72.6) |

[a]Reference[5]

**Table 2 | Preclinical model parameters**

| Parameter | LZD (%RSE) | TBI-223 (%RSE) |
|---|---|---|
| Mono-therapy PK-PD parameters[a] | | |
| EC$_{50}$ (mg/L) | 2.87 (9.7) | 2.86 (1.1) |
| Emax (day$^{-1}$) | 0.999 (2.3) | 0.444 (0.9) |
| Rate to effect compartment (day$^{-1}$) | 6.44 (0.01) | 98.1 (0.1) |
| Gamma | 0.215 (3.12) | 0.623 (2.9) |
| Protein binding (fu$_{human}$/fu$_{mouse}$) | 0.790/0.805[b] | 0.634/0.671 |
| Combo-therapy PK-PD parameters | | |
| EC$_{50}$ (mg/L) | 0.200 (115) | 0.155 (112) |
| Emax (day$^{-1}$) | 0.311(19) | 0.230(22) |
| Rate to effect compartment (day$^{-1}$) | 15.8(60) | 13.0(63) |
| Gamma | 1 FIX | 1 FIX |
| Potency targets | | |
| Minimum inhibitory concentration, MIC (mg/L) | 0.5 to 1 | 2 to 4 |
| Minimum bactericidal concentration, MBC$_{90}$ (mg/L) | 3.4 to 5 | 4 to 30 |
| Macrophage IC$_{90}$, (mg/L) | 6.75 | 4.2 |
| Caseum minimum bactericidal concentration, casMBC$_{90}$ (mg/L) | 43 | 46.8 |
| Preclinical safety testing[c] | | |
| Mammalian mitochondrial protein synthesis | 8 µM | >74 µM |
| [d]Safety margin = AUC$_{Rat-NOAEL}$/AUC$_{human}$ | 600 mg QD: 0.689 fold | 2400 mg QD: 16.4 fold |
| Rabbit lesion parameters | | |
| Ratio normal | 1.04 (12) | 1.01 (16) |
| Rate cellular | 10 FIX | 10 FIX |
| Ratio cellular | 0.977 (14) | 1.12 (21) |
| Rate outer cellular | 10 FIX | 10 FIX |
| Ratio outer cellular | 0.994 (15) | 1.14 (12) |
| Rate inner cellular | 10 FIX | 10 FIX |
| Ratio inner cellular | 0.957 (24) | 1.3 (3.0) |
| Rate outer caseum | 10 FIX | 1.72 (1.0) |
| Ratio outer caseum | 1.05 (7.0) | 1.49 (10) |
| Rate inner caseum | 10 FIX | 0.865 (2.0) |
| Ratio inner caseum | 1.03 (3.0) | 1.74 (14) |

[a]Based on the acute mouse model experiment.
[b]Reference[31]
[c]Reference[13]
[d]Based on 28-day rat toxicity study.

predictions. This could be due to the predicted human exposures being well within the experimental exposures in the animal efficacy experiments, Fig. S9. Additionally, bedaquiline and pretomanid doses in mice were selected to match human clinical exposures which likely also contributed to improved translation. In the future, when predicting outcomes of combination therapy with multiple drugs that have not yet had clinical outcome readouts, a more nuanced approach like the GPDI method might be necessary. However, we found that mouse experiments using the subacute infection model that only changed the dose of one compound in a combination were sufficient to predict clinical outcomes and could be used as a simplified approach for future experiments when investigating a novel single drug added to a combination.

Achieving adequate concentrations of TB drugs at the site of infection is important to treatment success[21,22]. To assess lesion coverage, we used a rabbit model due to its ability to predict clinical lesion penetration[1,18]. TBI-223 had similar lesion penetration parameters compared to linezolid, meaning the physiochemical properties that enable distribution into cellular and caseous environments were preserved. However, due to the higher systemic clearance of 32.8 L/h for TBI-223, compared to 7.9 L/h for linezolid, TBI-223 is eliminated faster than linezolid and even at higher doses its coverage in cellular lesions is roughly half that of linezolid. Due to the similar penetration parameters and a penetration close to one for both study drugs, the final efficacy model was reduced to have only plasma concentrations drive efficacy. While the comparable lesion penetration of both compounds simplified our analysis, this similarity further implies that the clinical performance of TBI-223 should align closely with linezolid.

Limitations of this study include extrapolation of TBI-223 doses beyond what was tested in healthy volunteers (1800 mg for SR and 2000 mg for IR). The absorption kinetics at these higher doses could be different than expected. However, saturation of the elimination kinetics was accounted for using a non-linear clearance model. Similarly, simulations that predicted outcomes for a co-formulation approach could also be less accurate due to possible changes in absorption kinetics when dosed together. The original clinical lesion coverage model for linezolid was limited to sparse PK sampling and the rate of penetration for rabbit lesions could not be predicted and compared. For the rabbit experiments human plasma linezolid exposure was matched in the rabbits, but the same was not done for TBI-223 which had twice the level of plasma exposure in rabbits compared to the clinical 2400 mg dose. Potentially the rate and ratio of penetration could be dose-dependent leading to different overall coverage in a clinical setting. Adjustments for inter-species differences in plasma protein binding were used for translation of mouse PK-PD data while

the macrophage IC$_{90}$ and caseum MBC assays used to obtain lesion-specific targets were not adjusted for protein binding, as they are performed in matrices that reproduce in vivo drug binding at the site of infection.

In conclusion, we provide dose selection recommendations for TBI-223, a potentially safer oxazolidinone than linezolid. In combination with bedaquiline and pretomanid, a 2400 mg daily dose of TBI-223 should provide optimal outcomes and has the potential to provide comparable efficacy to the linezolid-containing BPaL regimen. If no dose interruptions are necessary, we predict that TBI-223 may provide more favorable efficacy outcomes when compared to linezolid in combination with bedaquiline and pretomanid. Based on this comprehensive assessment, TBI-223 in combination with bedaquiline and pretomanid has been accelerated into combination Phase 2 trials without needing a monotherapy Phase 2A clinical trial. Furthermore, the end-to-end preclinical-clinical translation approach used here can be expected to provide improved dose selection and outcomes for future clinical trials of new tuberculosis drug candidates.

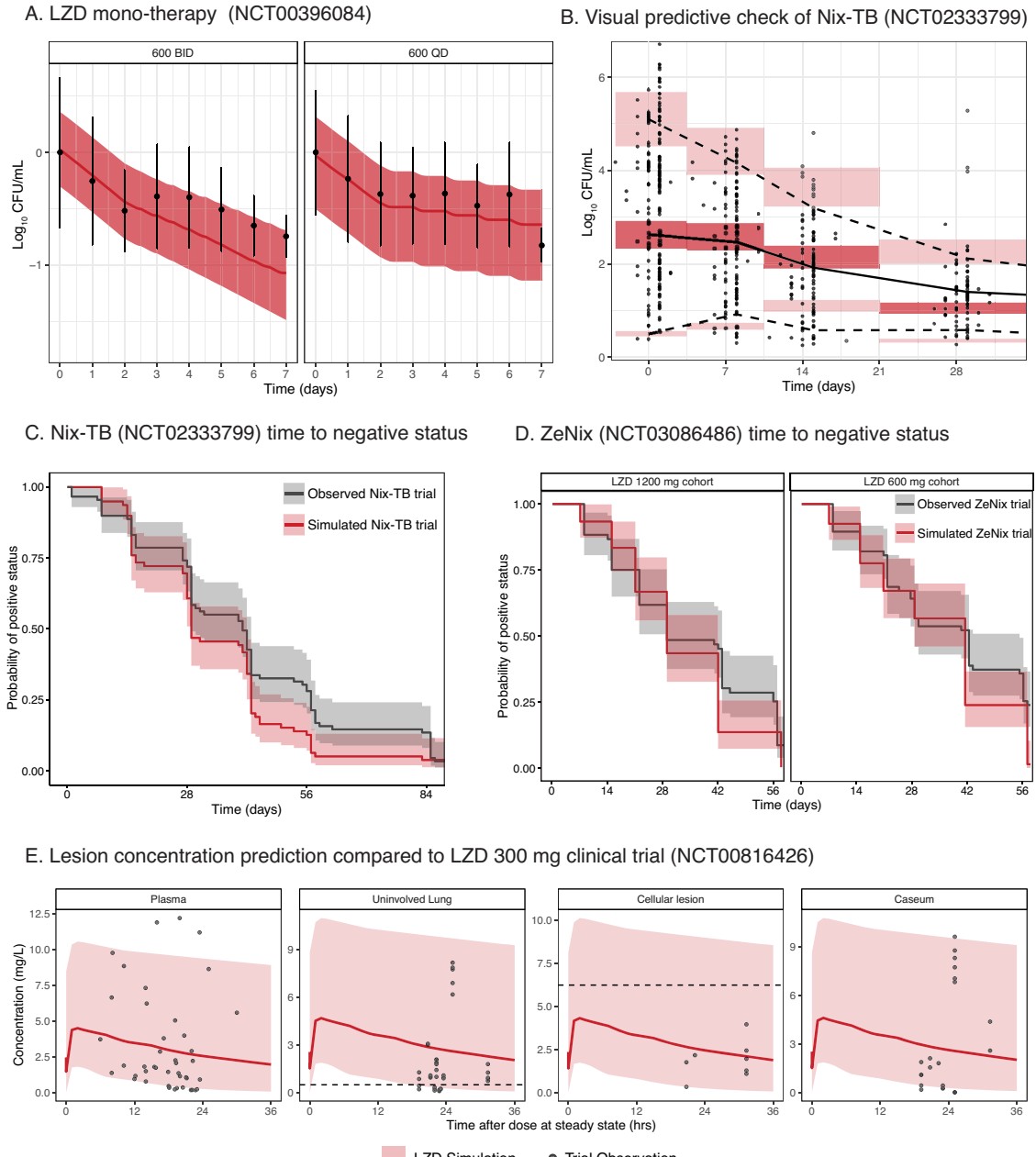

**Fig. 3 | Prediction of linezolid clinical endpoints compared to observed clinical data.** In each panel simulation data from the translational tool and clinical observed data are shown in red and black, respectively. Shaded areas show 95 percentile intervals and solid lines represent means. Pharmacokinetic (PK) concentrations were simulated using a published population PK model[5]. **A** Prediction of monotherapy CFU decreases as compared to phase 2a trial data where mean CFU change and standard deviations are plotted[17]. The 600 QD group consisted of $n = 10$ patients, and the 600 BID group had $n = 9$ patients. **B** Simulation of $Log_{10}$ CFU over time and compared to Nix-TB collected TTP data that was converted to $Log_{10}$ CFU/mL[2,28]. **C** Prediction of the probability of negative culture status over time on treatment compared to data from Nix-TB trial. Nix-TB trial predictions utilized

individual-level data and included patient dosing history, baseline TTP, and individual PK parameters. **D** Prediction of the probability of negative culture status over time on treatment in ZeNix trial[4]. Predictions included patient-level baseline TTP data and Kaplan–Meier plots are separated by dose (1200 mg once daily vs 600 mg once daily) and the mean baseline TTP value of 13 days. **E** Lesion distribution using parameters estimated in rabbits[2]. The selected monotherapy PD targets for LZD represented as dashed lines were MIC for uninvolved lung (0.5 mg/L), macrophage $IC_{90}$ for cellular lesions (6.25 mg/L), and caseum MBC90 for caseum (43 mg/L, outside y-axis range). Plasma is not considered a site of infection for tuberculosis and is therefore not included. CFU colony forming units, LZD linezolid, TTP time to positivity. Source data are provided as a Source Data file.

## Methods

All research complied with relevant ethical regulations and was approved by the respective institution including the Institutional Animal Care and Use Committee of BioDura Inc., Institutional Animal Care and Use Committee of Johns Hopkins University, Baltimore, MD, and Institutional Animal Care and Use Committee of the Center for Discovery and Innovation, Hackensack Meridian Health, Nutley, NJ.

### Pharmacokinetic experiments

Comprehensive toxicology studies in mice, rats, and dogs were performed for TBI-223 and linezolid as part of the initial IND filing. Pharmacokinetic data were collected during the experiments which were used for a multispecies pharmacokinetic model that predicted human projected concentrations for the Phase 1 trial of TBI-223. Appendix A of the supporting information describes this multispecies pharmacokinetic model in more detail. This pharmacokinetic model

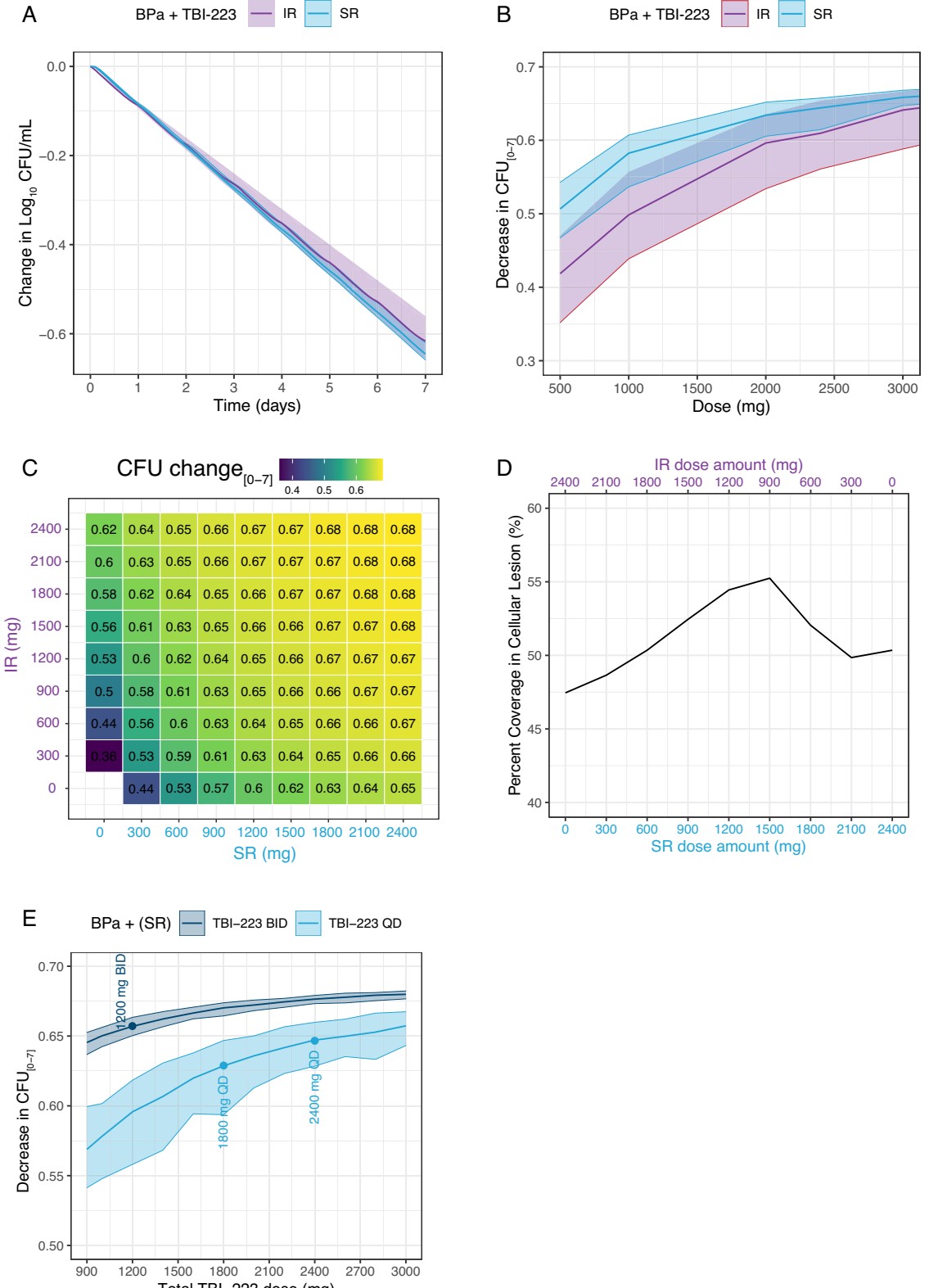

Fig. 4 | **Dose selection and formulation strategy for TBI-223.** Shaded areas show 95 percentile intervals and solid lines represent means. **A** Combination therapy prediction utilizing 2400 mg TBI-223 as either IR (purple) or SR (blue) formulation. **B** Dose-response of IR vs SR formulation. **C** Combination matrix of IR + SR with color and text in tiles showing the reduction in CFU for each dose over 7 days. **D** Percentage lesion coverage, ie, % of dosing interval in which intracellular concentration exceeds macrophage $IC_{90}$, in cellular lesions of additive IR and SR combination equal to 2400 mg. **E** Dose-response comparing once daily (navy) vs twice daily (light blue) dosing. IR immediate release, SR sustained release, $IC_{90}$ Inhibitory concentration 90%.

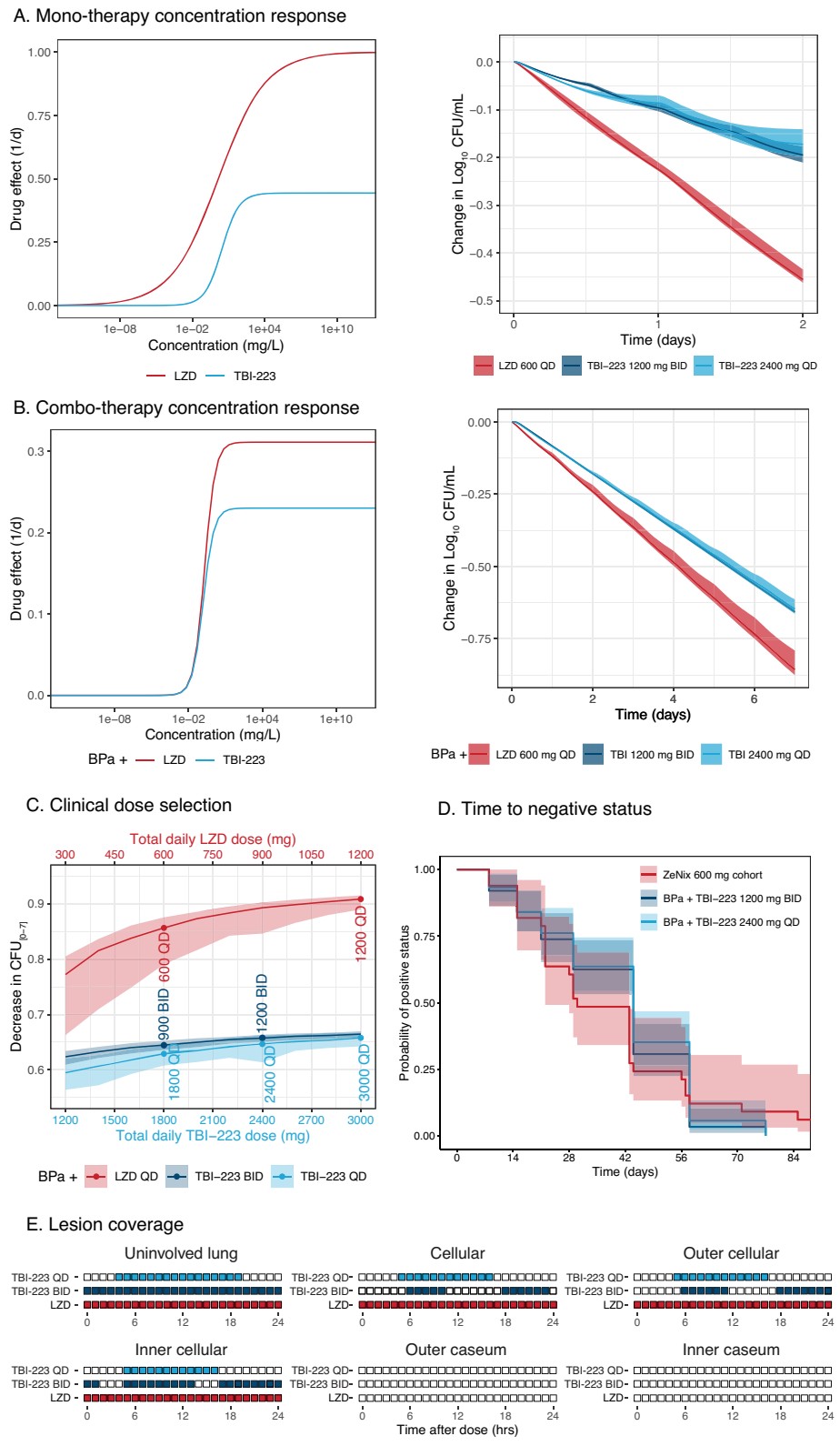

A. Mono-therapy concentration response

B. Combo-therapy concentration response

C. Clinical dose selection

D. Time to negative status

E. Lesion coverage

was repurposed to predict mouse pharmacokinetics for the PD studies described below.

Pre-clinical PK data were obtained from uninfected mice, rats, and beagle dogs in experiments performed at BioDuro, Inc. For the animal procedures, male C57BL/6 and BALB/c mice were purchased from Lingchang Biotechnology Co., Ltd (Shanghai, China), Sprague–Dawley rats from Beijing Vital River laboratories, and beagle dogs from Beijing

Marshall Biotechnology., Ltd. All animals were housed in cages with food and water available optionally. Laboratory conditions included a clean environment at 20–25 °C under 50–60% humidity and 12 h/12 h light/dark cycles. All the procedures involving animals were approved by the Institutional Animal Care and Use Committee (IACUC). PK results showed no sex differences for linezolid or TBI-223.

**Fig. 5 | TBI-223 clinical translation, comparison to linezolid and dose selection.** Shaded areas show 95 percentile intervals and solid lines represent means. Each panel compares linezolid (red) to TBI-223 administered either daily (light blue) or twice daily (navy). **A** Comparison of monotherapy PK-PD model shown by concentration-response and predicted decrease in CFU. **B** Combination PK-PD results including concentration-response and predicted change in CFU from baseline over seven days. **C** Dose-response of LZD and TBI-223 with top and bottom axis aligned to compare suggested low and high doses of linezolid and TBI-223. **D** Predicted time to culture conversion defined by bacterial compartment reaching <1 bacteria for TBI-223 without dose adjustments and real-world ZeNix data

(600 mg QD arm). **E** Lesion coverage is defined as lesion-specific drug concentrations above the respective lesion target concentration for each hour, Fig. S8. Each colored square represents 1 h that a drug is above its respective PD target. Empty blocks show drug concentration below the relevant monotherapy PD target and blue and navy blocks represent hours above target concentration for TBI-223 2400 mg daily and 1200 mg twice daily respectively and red blocks for linezolid (LZD). The selected monotherapy PD targets for TBI-223 were MIC for uninvolved lung (2 mg/L), macrophage $IC_{90}$ for cellular lesions (4.2 mg/L), and caseum $MBC_{90}$ for caseum (46.8 mg/L), Table 2: Potency targets.

Fed female BALB/c mice weighing ~25 g received single linezolid or TBI-223 doses of 5 mg/kg by tail vein injection (IV) or 100, 250, or 500 mg/kg by oral administration (gavage). C57BL/6 mice weighing approximately 30 g received TBI-223 5 mg/kg by tail vein injection or 100 mg/kg by gavage. Sprague–Dawley rats weighing ~250 g received linezolid 3 mg/kg by tail vein injection or 100 mg/kg by gavage. Beagle dogs weighing ~10 kg received linezolid or TBI-223 doses of 3 mg/kg by tail vein injection (IV) or 25 mg/kg by gavage or TBI-223 100, 250, or 500 mg/kg by gavage. Plasma was sampled at 5 min (IV only), 0.25, 0.5, 1, 2, 4, 7 and 24 h post-dose. Linezolid and TBI-223 were quantified by high-pressure liquid chromatography coupled to tandem mass spectrometry (LC/MS-MS). Protein precipitation extraction (PPT) was performed by adding acetonitrile (ACN) containing internal standard (IS, terfenadine) to 1 volume of plasma. The PPT mixtures were vortexed for 1 min and centrifuged at $3200 \times g$ for 15 min. The supernatant was then transferred for LC/MS-MS analysis.

The LC/MS-MS analysis was performed on a Sciex Applied Biosystems API 4000 triple-quadrupole mass spectrometer coupled to a Shimadzu LC-20AD to quantify the samples. Chromatography was conducted using an Agilent Kinetex 2.6 µm C18 100 A column (3 × 50 mm) under a reverse-phase gradient elution. The aqueous mobile phase (A) comprised 0.1% formic acid in Milli-Q deionized water and the organic mobile phase (B) contained 0.1% formic acid in ACN. The gradient was initiated with 8:2 for A to B at a flow rate of 0.8 mL/min, maintained until 0.30 min. From 0.30 to 1.50 min, the percentage of A was decreased to 10%. This composition was held constant until 2.00 min. Immediately after, the gradient was reverted back to 8:2 for 1 min to re-equilibrate the column. Multiple-reaction monitoring (MRM) of parent/daughter transitions in electrospray positive-ionization mode was used to quantify the analytes. The following MRM transitions were used respectively for linezolid (338.20/296.00), TBI-223 (366.27/296.10), and terfenadine (472.40/436.40). Sample analysis was accepted if the concentrations of the quality control samples were within 20% of the nominal concentration. Data processing was performed using Analyst software (version 1.5.1; Applied Biosystems Sciex).

### Pharmacodynamic experiments

**Mouse infection and drug administration.** All mouse infection experiments were performed in Biosafety Level 3 facilities using procedures approved by the Institutional Animal Care and Use Committee of Johns Hopkins University and used humane endpoints. Linezolid monotherapy PK-PD data from acute and chronic infection models in BALB/c mice were available from a previously published dose fractionation study[16]. A TBI-223 dose fractionation experiment was performed at the same time and under the same conditions as the linezolid. The monotherapy TBI-223 and linezolid dose fractionation experiment used the acute infection model with a high-dose aerosol infection and an incubation period of 6 days. Three total (cumulative) weekly doses of TBI-223 were used: 300, 1000, and 3000 mg/kg per week for 4 weeks with five mice per time point. Each total weekly dose was fractionated 4 ways: twice daily (BID) 7 days per week, BID 5 days per week (Mon–Fri), QD 5 days per week, and QD 3 days per week (Mon, Wed, Fri). Lung CFU counts were assessed after 4 weeks of

treatment. Combination PK-PD data of TBI-223 and linezolid administered with bedaquiline 25 mg/kg QD and pretomanid 25 mg/kg twice daily (BID) were collected from at least 6-week-old female BALB/c mice (Charles River Laboratories) in a high-dose aerosol infection model with a 17-day incubation period between infection and the start of treatment (Day 0). Four doses selected by expected clinical exposures and dose-response were tested for each oxazolidinone (12.5, 25, and 50 mg/kg BID and 100 mg/kg QD for linezolid and 15, 30, and 45 mg/kg BID and 100 mg/kg QD for TBI-223). Lung CFU counts were determined on Day 0 and after 4 and 8 weeks of treatment. All infections used the H37Rv strain of *M. tuberculosis*. To quantify bacterial infection, mice were sacrificed 3 days after the last drug dose to reduce the risk of drug carryover. Lungs were collected and homogenized in glass grinders at pre-specified time points during and after drug treatment. The homogenates were serially diluted in PBS and plated on Middlebrook 7H11 agar plates supplemented with 10% (v/v) OADC (GIBCO) and cycloheximide [10 mg/mL], carbenicillin [50 mg/mL], polymixin B [25 mg/mL] and trimethoprim [20 mg/mL], except that media used in the acute infection model studies did not contain selective antibiotics. Homogenates from mice treated with drug combinations were plated on the same agar media but with the addition of activated charcoal powder (0.4%w/v) to further prevent drug carryover. Colonies were counted after 4 and 6 weeks of incubation at 37 °C to ensure all cultivable bacteria would be detected.

### Rabbit lesion experiments

**Rabbit infection and drug administration.** Rabbits were selected for the lesion experiments for their consistency in developing lung cavities that mirror the complex pathology of human pulmonary cavitary tuberculosis[22]. All rabbit infection and drug administration experiments were performed in Biosafety Level 2 and 3 facilities and approved by the Institutional Animal Care and Use Committee of the Center for Discovery and Innovation, Hackensack Meridian Health, Nutley, NJ. Female New Zealand White rabbits (Charles River Laboratories), aged 12–14 weeks and weighing 2.5–2.8 kg, were maintained under specific pathogen-free conditions, and fed water and chow ad libitum. TBI-223 was obtained from the TB Alliance. In dose-finding studies, non-infected rabbits received a single oral dose of linezolid at 50 and 100 mg/kg, or TBI-223 at 50 and 150 mg/kg, both formulated in 0.5% carboxymethyl cellulose/0.5% Tween 80. Blood was collected at serial time points post-dose. Linezolid and TBI-223 were quantified in plasma as described below. To provide the human-equivalent dose in rabbits, plasma PK models were built using the initial PK data from these experiments and doses calculated that would provide similar rabbit drug exposures compared to human exposures. The human-equivalent efficacious dose was estimated to be 90 mg/kg for linezolid (equivalent to 1200 mg QD) and 200 mg/kg for TBI-223 (equivalent to 2400 mg QD). These doses were used thereafter in rabbits infected with TB 12–16 weeks before dosing. Rabbit infection, blood sampling, tissue dissection, and lesion processing followed previously described methods and used eight rabbits per experiment[23].

**Whole tissue and plasma drug quantification.** Linezolid concentrations in plasma and tissues were determined with stably labeled

linezolid-d3 used as internal standard. Neat 1 mg/mL DMSO stocks were serially diluted in 50/50 ACN /Milli-Q water for standard curves and quality control (QC) samples. Twenty μL of spiking solutions was added to 180 μL of drug-free plasma or control tissue homogenate to create standards and QCs. PPT was performed by adding 10 volumes of extraction solvent (1:1 ACN: methanol (MeOH)) containing 200 ng/mL internal standard linezolid-d3 to 1 volume of plasma or homogenized tissue sample. The mixtures were vortexed for 5 min and centrifuged at $3500 \times g$ for 5 min. The supernatant was then transferred for high-pressure liquid chromatography coupled to tandem mass spectrometry (LC/MS/MS) analysis.

LC/MS-MS analysis was performed on a Sciex Applied Biosystems Qtrap 6500+ triple-quadrupole mass spectrometer coupled to a Shimadzu Nexera X2 HPLC to quantify the clinical samples. Chromatography was performed with an Agilent Zorbax SB-C8 column (2.1 × 30 mm; particle size, 3.5 μm) using a reverse phase gradient elution. Gradients used 0.1% formic acid in Milli-Q deionized water for the aqueous mobile phase and 0.1% formic acid in ACN for the organic mobile phase. MRM of parent/daughter transitions in using APCI positive-ionization mode was used to detect the analytes. The following MRM transitions were used, respectively, for linezolid (338.2/ 296.1), linezolid-d3 (341.20/297.20). Sample analysis was accepted if the concentrations of the quality control samples were within 20% of the nominal concentration. Data processing was performed using Analyst software (version 1.6.2; Applied Biosystems Sciex). TBI-223 was quantified by LC/MS-MS as follows. Neat 1 mg/mL DMSO stocks of TBI-223 were serially diluted in 50/50 ACN/H$_2$O to create standard curves and quality control neat spiking solutions. Rabbit lung tissues and drug-free control rabbit lung were weighed and homogenized in 10 volumes of PBS. Homogenization was achieved using a FastPrep-24 instrument (MP Biomedicals) and 1.4 mm zirconium oxide beads (Bertin Corp.). 20 μL of neat spiking solutions were added to 180 μL of drug-free plasma or control tissue homogenate to create standards and quality control (QC) samples. Drug-free New Zealand White Rabbit K$_2$-EDTA plasma from BioIVT was used to build standard curves. Drug-free rabbit lungs were collected in-house. PPT was performed by adding 10 volumes of solvent containing internal standard (IS) to 1 volume of plasma or homogenized tissue sample. 1:1 ACN: Methanol (MeOH) containing 20 ng/mL Verapamil IS was added as the PPT solvent. The PPT mixtures were vortexed for 5 min and centrifuged at $3500 \times g$ for 5 min. The supernatant was then transferred for LC/MS-MS analysis.

LC/MS-MS analysis was performed on a Sciex Applied Biosystems Qtrap 6500+ triple-quadrupole mass spectrometer coupled to a Shimadzu Nexera X2 HPLC to quantify the clinical samples. Chromatography was performed with an Agilent Zorbax SB-C8 column (2.1 × 30 mm; particle size, 3.5 μm) using a reverse phase gradient elution. Gradients used 0.1% formic acid in Milli-Q deionized water for the aqueous mobile phase and 0.1% formic acid in ACN for the organic mobile phase. MRM of parent/daughter transitions in electrospray positive-ionization mode was used to quantify the analytes. The following MRM transitions were used respectively for TBI-223 (366.05/ 296.00) and verapamil (455.40/165.20). Sample analysis was accepted if the concentrations of the quality control samples were within 20% of the nominal concentration. Data processing was performed using Analyst software (version 1.6.2; Applied Biosystems Sciex).

**Laser-capture microdissection of rabbit lesion sections.** Twenty-five μm thick tissue sections were cut from infected rabbit lung biopsies using a Leica CM 1860UV (Buffalo Grove, IL) and thaw-mounted onto 1.4 μm thick Leica PET-Membrane FrameSlides (Buffalo Grove, IL) for laser capture microdissection. Tissue sections were immediately stored in sealed containers at −80 °C. Adjacent 10 μm thick tissue sections were thaw-mounted onto standard glass microscopy slides for H&E staining.

Cellular, necrotic (caseum), and uninvolved lung lesion areas totaling 3 million μm$^2$ were dissected from between 3 and 5 serial lung biopsy tissue sections using a Leica LMD6 system (Buffalo Grove, IL). Areas of cellular and caseous lesions were identified optically from the brightfield image scan and by comparison to the adjacent H&E reference tissue. Pooled dissected lesion tissues were collected into 0.25 mL standard PCR tubes and immediately transferred to −80 °C. On the day of analysis, 2 μL of PBS, 10 μL of 50/50 ACN/MilliQ water, and 50 μL of extraction solution (ACN/MeOH (1/1) with 1 ng/mL verapamil) was added to each tube, which was then vortexed for 5 min and centrifuged at 11,000 × $g$ for 5 min at room temperature. 50 μL of supernatant was transferred for LC/MS-MS analysis and diluted with an additional 50 μL of MilliQ water. Neat 1 mg/mL DMSO stocks for TBI-223 were diluted serially in 50/50 MeCN/MilliQ water to create standard curves and quality control spiking solutions. 10 μL of neat spiking solutions were added to 2 μL of lesion homogenate and extraction was performed by adding 50 μL of extraction solution ACN/MeOH (1/1) with 1 ng/mL verapamil. Extracts were vortexed for 5 min and centrifuged at 11,000 × $g$ for 5 min. 50 μL of supernatant was transferred for LC/MS-MS analysis and diluted with an additional 50 μL of MilliQ water. The total tissue volume of each pooled sample was determined based on the surface area of the pooled sections and the 25 μm tissue thickness. A dilution factor was used to normalize the tissue volumes with the standard curve for quantification.

**Antimycobacterial and lesion-specific concentration targets.** For the intracellular macrophage assay, THP-1 monocytes, cultured in RPMI 1640 supplemented with 10% fetal bovine serum and 2 mM l-glutamine, were seeded in 24-well plates and differentiated into macrophages using 100 nM phorbol 12-myristate 13-acetate (PMA). Post-differentiation, the macrophages were infected with the Erdman strain of M. tuberculosis at a multiplicity of infection (MOI) of 1:1. Following a 4 h infection period, extracellular bacteria were removed by washing with PBmS, and cells were treated with various concentrations of the test drugs. After 72 h, macrophages were lysed with 0.05% sodium dodecyl sulfate, and the lysates were plated on Middlebrook 7H11 agar to count colony-forming units (CFUs)[24].

For the ex vivo caseum assay, caseum was extracted from the lungs of *Mycobacterium tuberculosis*-infected rabbits and homogenized. The homogenates were then exposed to drug concentrations ranging from 0.01 to 512 μM. After a 7-day incubation at 37 °C, the treated caseum was serially diluted and plated on Middlebrook 7H11 agar. The minimum bactericidal concentration (MBC90), defined as the concentration needed to achieve a 90% reduction in bacterial burden, was determined by counting the CFUs on the agar plates[25].

### Modeling and simulation

**General modeling workflow.** The general workflow of modeling and simulation for the study is shown in Fig. 1. After data collation, a preclinical PK model was first established to link time-dependent concentration to response data that included lesion site of action concentrations and separately bacterial load in the mice efficacy models. An immune model was first fit to the baseline data to capture mouse immune response that decreases bacterial growth[26]. By estimating the true drug response devoid of immune effects, our PK-PD modeling employed parameters derived under the premise that identical free drug concentrations would yield equivalent PK-PD responses in both mice and humans[26,27]. For rabbit lesion distribution it was assumed that the estimated rate and ratio of penetration parameters that are linked to rabbit plasma concentrations are translatable to humans when clinical concentrations are accounted for[22].

After the models were built to establish concentration response, simulations using clinical concentrations were performed. Linezolid modeling used a two-compartment with non-linear clearance population PK model[5] while TBI-223 modeling required a one-compartment

model with non-linear clearance based on available Phase 1 data (Fig S1). Baseline TTP values for Nix-TB and ZeNix subjects were converted to CFU using a previously published and validated equation[28]. Censored events were removed and for culture conversion simulations, culture conversion was assumed to be reached at the moment the bacterial compartment reached 1 bacterium.

All analyses were conducted using NONMEM version 7.4 (ICON, plc.), Perl speaks NONMEM (PsN), R version 3.5.2 (R Foundation for Statistical Computing, Vienna, Austria). The xpose4 and ggplot2 R packages were utilized for model diagnostics and data visualization. The first-order conditional estimation with interaction method (FOCE + I) was used. Final models were selected based on parsimony, diagnostic visual predictive checks, and statistical significance based on the −2 log-likelihood change for inclusion of additional parameters in nested models, with a decrease of 3.84 points considered statistically different at the 0.05 level.

Clinical, rabbit, and mouse plasma PK parameter estimation and model building included linear vs non-linear clearance, first-order absorption, a lag time in absorption, and transit compartment models of absorption and multi-compartments of drug distribution.

**Mouse PK-PD.** To estimate drug response compared to untreated growth and death of bacterial burden in mice, a previously reported model to quantify the mouse immune response to *M. tuberculosis* infection in BALB/c mice was followed[26,27]. The model describes the natural growth and death of bacteria and an immune response that adjusts bacterial growth based on the duration of infection as the immune system requires time to come into full effect, and the size of initial infection where a larger infection elicits a faster immune response. Equations (1) and (2) show the relationship of these variables. To account for drug effect we tested linear, $EC_{50}$, $E_{max}$, and sigmoidal relationships as a direct effect on the bacterial compartment, Eq. (3). It was found that an effect compartmental model was necessary to best fit the data, Eq. (4).

$$\frac{dB}{dt} = K_g \times B \times \left(1 - \frac{K_B \times B^{\gamma_B}}{B_{50}^{\gamma_B} + B^{\gamma_B}}\right) \times \left(1 - \frac{K_T \times t^{\gamma_T}}{T_{50}^{\gamma_T} + t^{\gamma_T}}\right) - K_d \times B \quad (1)$$

$$\frac{dB}{dt} = K_g \times B \times \left(1 - \frac{K_B \times B^{\gamma_B}}{B_{50}^{\gamma_B} + B^{\gamma_B}}\right) \times \left(1 - \frac{K_T \times t^{\gamma_T}}{T_{50}^{\gamma_T} + t^{\gamma_T}}\right) - K_d \times B - EFF \times B \quad (2)$$

$$EFF = \frac{A_{effect}^{\gamma} \times E_{max}}{EC_{50}^{\gamma} + A_{effect}^{\gamma}} \quad (3)$$

$$\frac{dA_{effect}}{dt} = K_{effect} \times \left(\frac{A_{plasma}}{V_{plasma}} - A_{effect}\right) \quad (4)$$

$B$: bacterial number, $t$: incubation time since inoculation, $K_g$: bacterial growth rate, $K_d$: bacterial natural death rate, $K_B$: bacterial number-dependent maximal adaptive immune effect, $B_{50}$: the bacterial number that results in half of $K_B$, $\gamma_B$: steepness of bacterial number-dependent immune effect relationship, $K_T$: incubation time-dependent maximal adaptive immune effect, $T_{50}$: the bacterial number that results half of $K_T$, $\gamma_T$: steepness of time-dependent immune effect relationship, EFF: bacterial killing rate, $A_{effect}$: the concentration level associated with drug effect, $K_{effect}$: the rate of drug from the plasma compartment to the compartment associated with drug effect, $E_{max}$: the maximal level of drug effect, $EC_{50}$: the effect concentration that results in half of the maximal drug effect, $\gamma$: the steepness of the relationship between the drug concentration in the effect compartment and drug effect, $A_{plasma}$: the amount of drug in the central, observed plasma concentration compartment, $V_c$: the estimated volume of the plasma compartment.

For monotherapy PK-PD simulations efficacy parameters from the acute mouse model (6 days incubation with bacteria) were used to simulate the first phase of kill while the chronic mouse model (32 days of incubation with bacteria) was used to describe the second phase kill prediction for linezolid (Table S1) to account for the biphasic kill that could be due to population-level heterogeneity in persistence, intrinsic resistance expression or multiple other factors[1,29]. To estimate the effect of linezolid and TBI-223 in combination with bedaquiline and pretomanid, several methods were attempted that looked at direct vs indirect response, linear vs sigmoidal relationships between effect and concentration, adding bedaquiline and pretomanid monotherapy or estimating a fixed additive response for bedaquiline and pretomanid without oxazolidinone on board. The model that provided the most accurate fit for combination therapy was an effect compartment model, wherein efficacy was determined using a sigmoidal Emax model, to describe the relationship between drug concentration data and bacterial decline. Additionally, methods that estimated $EC_{50}$ with fixed $E_{max}$ parameters from monotherapy experiments, including acute, subacute, and chronic infection models were attempted as a way to estimate an $EC_{50}$ shift or correction factor for oxazolidinone when bedaquiline and pretomanid were added. However, this constraint did not provide reasonable fits, and $EC_{50}$ and $E_{max}$ were estimated independently without monotherapy priors.

**Rabbit lesion distribution.** Rabbit lesion PK modeling was in line with previously published methods[15,22,30]. After determining the best plasma PK model, individual plasma parameters were fixed for each rabbit, and lesion parameters were estimated for the population using Eq. (5).

$$\frac{dC_{lesion}}{dt} = Rate_{lesion} \times \left(Ratio_{lesion} \times \frac{A_{plasma}}{V_{plasma}} - C_{lesion}\right) \quad (5)$$

$C_{lesion}$ represents the drug concentration within uninvolved lung and lesion, $Rate_{lesion}$ is the inter-compartment rate constant for the transfer of drug from the plasma to lung or lesion, $Ratio_{lesion}$ is the penetration coefficient value between lung or lesion and plasma, and $A_{plasma}/V_{plasma}$ is the drug concentration in plasma at time $t$.

For lesions that only had single time points above the limit of quantification a rate of $10\,h^{-1}$ was assumed and only the ratio of penetration was estimated. Coverage was defined as the time lesion-specific concentration was above its respective lesion target. *Mycobacterium tuberculosis* populations residing in different tissue microenvironments are phenotypically distinct and respond differently to drug treatment and therefore antimicrobial targets that replicate these microenvironments were used. Specifically for cellular lesions the macrophage $IC_{90}$ values were used as the most appropriate target and caseous lesions used caseum $MBC_{90}$.

**Simulation.** Simulations were performed in R version 3.5.2 using the mlxR package (version 4.0.6) and used clinically estimated plasma parameters linked to the estimated response parameters from the preclinical model after appropriate adjustment for protein binding. EBA values were calculated as the daily change of CFU counts over specific days with treatment of linezolid and TBI-223. In the simulation, patients not receiving any drug treatment showed minimal changes in bacteria count in the first two days. Therefore, the model assumed no natural change in CFU during this initial period, attributing any changes solely to the effect of the drug. Equation (2) therefore simplifies to the Eq. (6) where $K_{net}$ equals 0.

$$\frac{dB}{dt} = K_{net} \times B - EFF \times B \quad (6)$$

Simulations were repeated 1000 times for predicting clinical studies conducted for each drug. For lesion coverage, no protein binding adjustments were made.

## Statistics and reproducibility

Source data and model code are available. The preclinical data generated in this study have been deposited in the Figshare database under accession code https://doi.org/10.6084/m9.figshare.26054098.v1. The model simulation code is provided in Appendix B of the supporting information. No data were excluded from model building. No statistical method was used to predetermine sample size. For animal experiments, the experiments were not randomized and investigators were not blinded to drug treatment and outcomes during experiments.

## Reporting summary

Further information on research design is available in the Nature Portfolio Reporting Summary linked to this article.

## Data availability

Source data are provided with this paper. The preclinical data generated in this study have been deposited in the Figshare database under accession code https://doi.org/10.6084/m9.figshare.26054098.v1. The model simulation code is provided in Appendix B of the supporting information. Source data are provided with this paper.

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

## Acknowledgements

We graciously thank Jerry Nedelman for his helpful discussions, input, and support during the design and execution of this study.

## Author contributions

N.S and R.S., V.D. and E.L.N. performed study concept and design; N.S. performed data analysis, model development and writing, review, and revision of the paper; J.E., M.I., B.P.S., Q.W. contributed to data handling and development of methodology, R.T., S.T., H.S., A.G., K.B. assisted in the design and performance of the mouse infection experiments. M.G., M.Z., M.X., and J.P.S. performed rabbit infection experiments and lesion quantification, and T.Y. provided technical and material support. All authors contributed to interpretation of the data and results, and read and approved the final paper.

## Competing interests

The authors declare no competing interests.
