## [Peer Review File · Nature Communications]

REVIEWER COMMENTS

Reviewer #1 (Remarks to the Author):

The study by Strydom et al provides a translational investigation of the novel TB drug candidate TBI-223 in comparison to linezolid. The authors parameterize their framework using preclinical data, validate it for linezolid and then use it to derive optimized dosing regimens for TBI-223. The study is timely and the approach sound. The conclusions seem overall justified. The paper is well written but lacks important details on the methodology used (see detailed comments below).

An overall comment is, that the value of the lesion PK data is not fully clear in this work. Lesion PK does not drive the PK-PD model and it seems not required within the model-based framework to derive the optimized dosing regimen. In favor of presenting the lesion PK in detail, the authors are encouraged to provide more details on the modelling and simulations workflows which are not fully transparent. I would highly encourage to enhance the manuscript with provision of model code, e.g. of the final clinical trial simulation.

Further comments in order of appearance of the manuscript:

I. 96: Unbound or total concentrations?

I. 285: Please provide composition of the gradient

I. 357: Free drug concentrations were assumed to be same between animals and humans: I do not see that free drug concentrations were experimentally determined. Please elaborate when free concentrations were used and how the underlying info on protein binding in plasma and tissues was determined.

I. 383: Rename “delayed effect model” to the more common description “effect compartment model”

I. 388: Please also explain A2 and V1.

I. 388: It is unclear at which stage the lesion concentrations are used in the PKPD model. To me, it looks like that total plasma concentrations (A2/V1) drive the effect. Please explain and expand this section to better guide the reader.

I. 406: Combo modelling strategy is quite unclear (even after reading the discussion). Please provide more details what is meant with “a direct sigmoidal model with delay and all parameters ...”

I. 413: Sentence seems incomplete.

I. 419: Lesion target concentration – provide a cross-reference to Figure S8 and give references for the reported lesion target concentrations in the Figure caption.

Fig 2: Monotherapy panel: Was Emax experimentally established for both linezolid and TBI-223? Interestingly, the authors estimate an Emax value of 0.999 /day for linezolid and 0.444 /day for TBI-223, while in the raw data, TBI-223 seems to be more effective. Moreover, the less effective linezolid gives a

better killing profile in combination as compared to TBI-223 in combination. This seems to be a bit counterintuitive to me. Please comment.

Fig S1: Facet IR: Data points continue to follow the monoexponential decline but model predictions remain constant after approx. 25 h. Is this just a binning issue? Please comment and resolve.

Reviewer #2 (Remarks to the Author):

This is a very thoroughly executed, well-written study on the potential of the oxazolidinone TBI-223 to improve the bedaquiline, pretomanid, linezolid (BPAL) regimen in the treatment of tuberculosis.

Given the toxicity that comes with linezolid, it is particularly relevant that the authors evaluated whether linezolid replacement for TBI-223, a potentially safer alternative, could retain the high efficacy of the BPAL regimen. The researchers took on the challenge of using a data-driven modeling approach, in which preclinical in vitro and in vivo data as well as data from clinical trials were combined to predict clinical activity of TBI-223 (both as single drug and in drug combinations) and lesion penetration with linezolid as comparator. Integration of international preclinical and clinical data obtained by various research groups in predictive models is extremely valuable as it leads to optimal use of data to guide the design of subsequent clinical trials. This is underlined by the fact that the bedaquiline, pretomanid, and TBI-223 combination has been accelerated into a phase 2 trial based on the data from this study, thereby omitting the need for phase 2A monotherapy trials. The most important finding generated by this approach is that TBI-223 dosed at 1200-2400 mg daily in combination with bedaquiline and pretomanid would be equally active as the BPAL regimen with >90% of patients predicted to reach culture conversion after 2 months of treatment.

Although the study is well-reasoned, there are several points of criticism. First, in contrast to the results section, the methods section seems quite incomplete and not harmonized. It is also not clear whether the experimental data were specifically gathered for this study, or if this study concerns re-use of data. Thirdly, title states "as a safer and effective therapy against tuberculosis". However, the data do not directly show that the proposed dose of TBI-223 in the BPAL regimen is indeed safer. This should be nuanced. Lastly, the contribution of the lesion penetration section to the overall relevance of the study is quite unclear. These points are, amongst others, more elaborately addressed in the comments below.

General remark

- Make sure to use either the term 'non-clinical' or 'pre-clinical' consistently throughout the manuscript.

Abstract

- "TBI-223 is a novel and more selective oxazolidinone...". What is meant by the term 'selective'. Is this the higher selectivity towards bacterial ribosomes instead of mitochondrial ribosomes? I suggest the authors either explain or remove this term.
- "... and describe its efficacy and safety profile as compared to BPaL" – did you mean to compare TBI-223 to the BPaL regimen, or to compare TBI-223 to linezolid? In the last paragraph of the introduction the same sentence is used, but there it says "compared to linezolid". Please clarify.
- The abstract is a nice summary of the work performed, but the direct aim of the study can be addressed more specifically.
- Minor comment: the term "...nonclinical monotherapy" is not quite correct, as a single agent is never considered as treatment of active tuberculosis. It would be better to address this as "pre-clinical mono-exposure".

Introduction

- "Linezolid dose reduction was necessary..." – it is interesting that the incidence of peripheral neuropathy and myelosuppression in the Nix-trial (81% and 48%, respectively) were much higher than in the ZeNix trial (38% and 22%, respectively) for the high dose 1200 mg 26 weeks arms. Are there any explanations known for this difference?
 - "Using pharmacokinetics and safety data ... the therapeutic window of linezolid is quite narrow" – The point being made here is that linezolid has a narrow therapeutic window. However, the way this is explained is quite confusing:
 - o Additional explanation is needed for the IC50 of 1.3 mg/L. Where was this value obtained? A reference would be in place here.
 - o "This is consistent with previous studies that showed mitochondrial toxicity and safety events were associated with trough levels and susceptibility breakpoints greater than 2 mg/L"
- I do not understand how toxicity events can be associated with susceptibility breakpoints. I could not find a statement about this association in reference (7). That reference does state that the AUC0-24 is linked to toxicity, but they also mention this could be because AUC0-24 is collinear with trough levels. Please clarify.

o “Considering that the minimum inhibitory concentration of linezolid to cover 50% and 90% of Mycobacterium tuberculosis clinical isolates is 0.5 mg/L and 1.0 mg/L, respectively (8)” – this is quite a firm statement, based on only one study. However, when I look in reference (8), I am not sure where these values were obtained. 0.5 and 1.0 mg/L seem to be the MIC50 and MIC90 for Nocardia brasiliensis. The paper mentions an MIC50 and MIC90 for linezolid of 1 and 2 mg/L, respectively, but refers to another paper (PMID 16189119). Were these values also used as model parameters (see Table 1: MIC 0.5-1.0 mg/L)? Please check whether the values are correct.

• “We used a data-driven approach ... to provide safer, more effective TB regimens” – was safety indeed an outcome parameter? It is not discussed as such in the results section.

• “Additionally, we believe new and existing drugs regimens should be systematically evaluated for lesion distribution as achieving adequate exposures at the site of infection is crucial for successful treatment” -- To be the devil’s advocate here – if treatment efficacy is already a primary outcome parameter, would it really be necessary to dive into the lesion distribution? I can imagine that when a certain drug regimen is less active than another it could be interesting to see whether this could be explained by a difference in tissue penetration, but what is the exact additive value of looking into tissue penetration in this study?

Methods

• Although I understand the authors have to comply to the word count restriction, quite a lot of useful information on the methodology of especially the animal experiments is missing. The authors refer to many other papers for the methods, but it would be good to give a brief summary so that the readers get a general idea of the methods used, instead of having to dive in to literature repeatedly. The LC/MS-MS methods on the other hand are described in much detail. Consider using the supplementary information for more in depth explanation on how the different studies were conducted. My comments below are in line with this general comment.

Mouse infection and drug administration.

• Useful information to add here would be (and this also accounts for the PK studies and rabbit studies):

o How were the doses of TBI-223 and linezolid chosen?

o How big were the group sizes?

o What route of infection was used in the acute infection model ?

o How much time was there in between infection and start of treatment?

o How were the CFU-counts determined? On middlebrook agar (7H10 or 7H11?) with OADC? Were the samples washed or were charcoal-containing plates used to prevent drug carry-over?

o Were humane endpoints considered?

• Was the MIC of TBI-223 and linezolid determined for the H37Rv? Or was the range of MIC (as indicated in Table 1) used?

- Were the pharmacodynamic-studies with BPaL really necessary to conduct, or could ‘historical data’ also have been used here?
- The authors mention ‘PK-PD data’, but were pharmacokinetics assessed in these experiments? It seems only PD was studied in these studies, as there is no information on blood sampling, time points, or plasma concentration determinations.

Plasma drug quantification

- Quite a lot of PK experiments were performed with various animals (mice, rats, beagle dogs). Were all these experiments really necessary for the aim of this study? Were all these data used as input for the predictive model? For example, why was PK after intravenous administration tested, as both linezolid and TBI-223 can be dosed orally in animals and humans?
- In line with this, considering the 3Rs of Reduction, Refinement, and Replacement, did the aim of the study truly justify the numbers of animals used and discomfort levels in these studies? Or was this experiment secondary to a primary study within which separate efficacy was assessed and described?

Rabbit lesion experiments

- Antimycobacterial and lesion-specific concentration targets
- o Please provide extra information on the rationale and methods used for these experiments to guide the readers.
- o Why are these studies placed within the ‘rabbit lesion experiments’-section?
- Rabbit infection and drug administration
- o Why is information provided on animal weight and housing conditions here, but not for the mouse drug activity studies and for the PK studies in mice, rats and beagles?
- o How was the human-equivalent efficacious dose calculated? ‘Based on modeling’ is quite blunt.
- o How was the human-equivalent efficacious dose for TBI-223 determined? Based on the clinical phase I data?
- o The human equivalent doses were determined in uninfected rabbits, while the tissue penetration experiments were executed in infected animals. Does infection influence the PK of linezolid and TBI-223?
- o How many rabbits were used in these experiments?
- o Why were rabbits chosen for these studies? Could using C3HeB/FeJ (Kramnik) mice also have been an option as they develop necrotizing granulomas as well?
- o Was drug activity not assessed in the rabbit studies?

General modelling workflow

- Although figure 1 depicts a comprehensive overview of the modelling approach that was used, the general set-up of the study is still quite difficult to understand. Was the model based on already performed animal experiments? Or were the animal experiments chosen such that clinical outcome can be optimally predicted? And in this case, how is decided what experiments need to be performed in order to be able to make predictions about TBI-223's clinical activity? A paragraph explaining this would better guide the readers through the study.
- "After data collation, a nonclinical PK model..." – typo in 'collation'; also, please clarify 'response data'.
- Two assumptions in the model are mentioned in the first paragraph (on the PK-PD response in mice versus humans and on the ratio of penetration parameters) – can this be substantiated with references?
- "For culture conversion simulations culture conversion was assumed to be reached at the moment the bacterial compartment reached 1 bacterium" – On what information is this assumption based? Would that mean that a patient's sputum culture would turn positive when there are 2 bacteria present in the lungs? This might be quite a strict assumption, as it does not match the technical sensitivity/lower limit of detection of mycobacterial cultures in practice.
- "Clinical, rabbit, mouse plasma PK estimation and model building included..." – does this mean that the PK information obtained in rats and beagles was not used in the modelling?
- "For monotherapy EBA simulations efficacy parameters ... intrinsic resistance expression or multiple other factors" – were chronic mouse model experiments only performed for linezolid, or also for TBI-223?
- "For cellular lesions the macrophage IC90 value was used as the most appropriate target and caseous lesions used caseum MBC90." – can you explain why these parameters were most appropriate best for each type of lesion?

Results

- In the first paragraph the authors refer to Appendix A – I think this should be Supplementary figure 1?
- "The experiments used an acute 6-day infection model ... to 1000 mg/kg for TBI-223" – this information seems more in place in the method section.
- Subhead 2: comparison of linezolid predictions with clinical endpoints
 - o The authors state twice that the model 'performed well'. Can you explain when you would say a model performs 'well'? As a reader with limited background in modelling it is difficult to estimate whether this can indeed be considered as a 'well-performing'-model.
 - o "observations were within range of the simulations" (Figure 3E) – am I correct that the range of the simulations is rather large?
- Subhead 4: dose recommendations for TBI-223 and its comparison to linezolid

o “To evaluate the treatment duration-shortening potential of TBI-223 in combination” – this is a tricky statement, as the authors did not look at the sterilizing activity of TBI-223 (or linezolid) alone or in combination. It is questionable if short term outcome parameters such as culture conversion can be used as a proxy for treatment shortening potential.

Figure 1:

- “collect dose ranging preclinical data from suitable animal models” – what makes an animal model suitable?
- In the ‘Clinical prediction’-panel – what is the difference between the red and the blue lines?
- Can you explain the ‘Lesion distribution and coverage’-figures? What are we looking at here?

Figure 2:

- The caption is very brief. Considering the broad public of Nature Communications, the authors might want to elaborate on these figures. I imagine a tremendous amount of work was put in all the animal studies, and these deserved to be well-understood.

Figure 4:

- Please explain the abbreviations: IR, SR, IC90.

Supporting information:

- Consider adding figure captions for S3-S6. These are difficult to interpret without explanation.

Discussion

- The authors have written a well-reasoned discussion that places their findings in a relevant broader context. It is very nice to read that this study omitted the need for phase 2A monotherapy studies.
- In the paragraph starting with “A 2400 mg dose of TBI-223 is predicted to produce concentrations...” the safety of TBI-223 is addressed, which is a highly relevant topic, since toxicity of linezolid is the main issue of the BPAL regimen as discussed elaborately in the introduction. However, it is not clear whether safety of TBI-223 was assessed within this study. Table 1 shows ‘safety margin’ as one of the model parameters, but was safety assessed in the experiments performed in the current study, or was this parameter based on a previously executed toxicity study in rats? The results section focusses on anti-tuberculosis activity of TBI-223 and lesion penetration, but toxicity of TBI-223 is not discussed. To understand whether the proposed dose of 2400 mg TBI-223 would actually be feasible in terms of toxicity, it would be good to elaborate on this topic.
- “we predict that TBI-223 will provide more favorable long-term safety and efficacy outcomes” -- ‘Long-term efficacy’ implies that the study looked at the ability of TBI-223 to cure TB without relapse. However, this study only addresses ‘short-term’ clinical outcomes such as EBA, time to culture conversion and time to positivity. The authors might want to rephrase this sentence and mention that ‘long-term’ outcome parameters such as cure without relapse of infection were not studied here. This can be seen as a

limitation of the study, since short-term clinical outcomes such as EBA do not necessarily predict relapse-free curation.

REVIEWER COMMENTS

Reviewer #1 (Remarks to the Author):

The study by Strydom et al provides a translational investigation of the novel TB drug candidate TBI-223 in comparison to linezolid. The authors parameterize their framework using preclinical data, validate it for linezolid and then use it to derive optimized dosing regimens for TBI-223. The study is timely and the approach sound. The conclusions seem overall justified. The paper is well written but lacks important details on the methodology used (see detailed comments below).

We thank the reviewer for acknowledging the merits of our study and for their constructive feedback on the need for more detailed methodological information. We have addressed each of the comments in detail below.

An overall comment is, that the value of the lesion PK data is not fully clear in this work. Lesion PK does not drive the PK-PD model and it seems not required within the model-based framework to derive the optimized dosing regimen. In favor of presenting the lesion PK in detail, the authors are encouraged to provide more details on the modelling and simulations workflows which are not fully transparent. I would highly encourage to enhance the manuscript with provision of model code, e.g. of the final clinical trial simulation.

We acknowledge the valid concern raised by both reviewers regarding the clarity of the lesion PK data's value and its role in our PK-PD model. To address this, we have revised the manuscript to explain the significance of lesion PK data in our study and enhanced the transparency of our modelling and simulation workflows. Additionally, in line with your suggestion, we have provided the model code for our clinical simulations, to offer comprehensive insight into our methods and facilitate understanding of our methods.

Further comments in order of appearance of the manuscript:

I. 96: Unbound or total concentrations?

Corrected to "Total drug concentrations"

I. 285: Please provide composition of the gradient

Corrected as follows:

"Chromatography was conducted using an Agilent Kinetex 2.6 μm C18 100A column (3 \times 50 mm) under a reverse-phase gradient elution. The aqueous mobile phase (A) comprised 0.1% formic acid in Milli-Q deionized water and the organic mobile phase (B) contained 0.1% formic acid in acetonitrile (ACN). The gradient was initiated with 8:2 A:B at a flow rate of 0.8 mL/min, maintained until 0.30 min. From 0.30 to 1.50 min, the percentage of A was decreased to 10%. This composition was held constant until 2.00 min. Immediately after, the gradient was reverted back to 8:2 for one minute to re-equilibrate the column"

I. 357: Free drug concentrations were assumed to be same between animals an humans: I do not see that free drug concentrations were experimentally determined. Please elaborate when free concentrations were used and how the underlying info on protein binding in plasma and tissues was determined.

We accounted for differences in protein binding between mice and humans, with the experimental values listed in Table 1 under the "Mono-therapy PK-PD parameters" section. These values were determined using equilibrium dialysis.

We've revised this sentence in question in the methods section as follows:

"By estimating the true drug response devoid of immune effects, our PK-PD modeling employed parameters derived under the premise that identical free drug concentrations would yield equivalent PK-PD responses in both mice and humans."

I. 383: Rename “delayed effect model” to the more common description “effect compartment model”

We have replaced "delayed effect model" with the standard term "effect compartment model" throughout the manuscript

I. 388: Please also explain A2 and V1.

In improving this section we have renamed A2 and V1 to A_{plasma} and V_c respectively and added the following: A_{plasma} : the amount of drug in the central, observed plasma compartment, V_c : the estimated volume of the plasma compartment.

I. 388: It is unclear at which stage the lesion concentrations are used in the PKPD model. To me, it looks like that total plasma concentrations (A2/V1) drive the effect. Please explain and expand this section to better guide the reader.

The reviewer is correct that only plasma concentrations drive the effect in our model. In this study, evaluating lesion distribution was crucial not to predict TBI-223's efficacy per se, but to discern if its tissue penetration could present an advantage or disadvantage compared to linezolid in a clinical scenario. Finding similar lesion penetration for both compounds streamlined our analysis, and ratios close to one suggested similar exposure levels in plasma and lesion which further allowed us to simplify our modeling approach. The similarities suggests that TBI-223's performance should be on par with linezolid, neither exceeding nor falling short in actual clinical settings. Additionally, our team is engaged in building comprehensive PK-PD models to determine inter- and intra-lesion efficacy, an extensive endeavor that falls beyond the scope of this current study. However, with similar exposures expected across lesions for both linezolid and TBI-223, it allowed us to reduce our assumptions to an apples-to-apples comparison using mouse PK-PD modeling to predict clinical outcomes.

We have expanded on this in the discussion to include more transparency on this topic:

“Due to the similar penetration parameters and a penetration close to one for both study drugs, the final efficacy model was reduced to have only plasma concentrations drive efficacy. While the comparable lesion penetration of both compounds simplified our analysis, this similarity further implies that the clinical performance of TBI-223 should align closely with linezolid.”

I. 406: Combo modelling strategy is quite unclear (even after reading the discussion). Please provide more details what is meant with “a direct sigmoidal model with delay and all parameters ...”

We acknowledge this text has incorrect phrasing. In the revised manuscript, we have clarified this aspect as follows:

"The model that provided the most accurate fit for combination therapy was an effect compartment model, wherein efficacy was determined using a sigmoidal E_{max} model, to describe the relationship between drug concentration data and bacterial decline. Additionally, methods that estimated EC_{50} with fixed E_{max} parameters

from monotherapy experiments, including acute, subacute and chronic infection models were attempted as a way to estimate an EC₅₀ shift or correction factor for oxazolidinone when bedaquiline and pretomanid were added. However, this constraint did not provide reasonable fits, and EC₅₀ and E_{max} were estimated independently without monotherapy priors."

I. 413: Sentence seems incomplete.

Corrected to "C_{lesion} represents the drug concentration within uninvolved lung and lesion"

I. 419: Lesion target concentration – provide a cross-reference to Figure S8 and give references for the reported lesion target concentrations in the Figure caption.

Corrected in the figure legend

Fig 2: Monotherapy panel: Was E_{max} experimentally established for both linezolid and TBI-223? Interestingly, the authors estimate an E_{max} value of 0.999 /day for linezolid and 0.444 /day for TBI-223, while in the raw data, TBI-223 seems to be more effective. Moreover, the less effective linezolid gives a better killing profile in combination as compared to TBI-223 in combination. This seems to be a bit counterintuitive to me. Please comment.

There may be a misunderstanding regarding the doses evaluated. In the Mono-therapy panel, only the grey and yellow lines can be directly compared. The LZD trials tested total weekly doses of 100, 300, and 1000 mg/kg, whereas for TBI-223, the doses were 300, 1000, and 3000 mg/kg. Looking at the 1000 mg/kg data, LZD reduced the bacterial load to less than 5 log, outperforming TBI-223, which achieved a reduction to 6 log at the same dose. Notably, 300 mg/kg of TBI-223 demonstrated a response similar to 100 mg/kg LZD, indicating a 3-fold difference. Therefore, considering the pharmacokinetic disparities, a 2.25-fold difference in E_{max} is in line with the observations.

Fig S1: Facet IR: Data points continue to follow the monoexponential decline but model predictions remain constant after approx. 25 h. Is this just a binning issue? Please comment and resolve.

Yes, this was a combination of BLQ's and binning. We changed to the least square method for binning and adjusted the x-axis.

Reviewer #2 (Remarks to the Author):

This is a very thoroughly executed, well-written study on the potential of the oxazolidinone TBI-223 to improve the bedaquiline, pretomanid, linezolid (BPaL) regimen in the treatment of tuberculosis.

Given the toxicity that comes with linezolid, it is particularly relevant that the authors evaluated whether linezolid replacement for TBI-223, a potentially safer alternative, could retain the high efficacy of the BPaL regimen. The researchers took on the challenge of using a data-driven modeling approach, in which preclinical in vitro and in vivo data as well as data from clinical trials were combined to predict clinical activity of TBI-223 (both as single drug and in drug combinations) and lesion penetration with linezolid as comparator. Integration of international preclinical and clinical data obtained by various research groups in predictive models is extremely valuable as it leads to optimal use of data to guide the design of subsequent clinical trials. This is underlined by the fact that the bedaquiline, pretomanid, and TBI-223 combination has been accelerated into a phase 2 trial based on the data from this study, thereby omitting the need for phase 2A monotherapy trials. The most important finding generated by this approach is that TBI-223 dosed at 1200-2400 mg daily in combination with bedaquiline and pretomanid would be equally active as the BPaL regimen with >90% of patients predicted to reach culture conversion after 2 months of treatment.

Although the study is well-reasoned, there are several points of criticism. First, in contrast to the results section, the methods section seems quite incomplete and not harmonized. It is also not clear whether the experimental data were specifically gathered for this study, or if this study concerns re-use of data. Thirdly, title states “as a safer and effective therapy against tuberculosis”. However, the data do not directly show that the proposed dose of TBI-223 in the BPaL regimen is indeed safer. This should be nuanced. Lastly, the contribution of the lesion penetration section to the overall relevance of the study is quite unclear. These points are, amongst others, more elaborately addressed in the comments below.

We sincerely thank the reviewer for recognizing the thoroughness of our work and the effort we have put into investigating the potential of TBI-223 to improve the BPaL regimen for tuberculosis treatment. We appreciate the constructive criticism and have addressed each point in our revisions to provide a more comprehensive and accurate presentation of our study.

Regarding the reviewer’s major feedback:

1. We acknowledge the overstatement about safety in our title. The initial choice of words was based on in vivo safety margins and long term clinical safety will be investigated. We have expanded the discussion to provide clarity on the justification of potential therapeutic benefit. In addition to the discussion on predicted safety margins we have added a statement on clinical safety from healthy volunteer studies: “The proposed doses demonstrated safety in short-term healthy volunteer studies and long-term safety will be determined in phase 2 trials”. Considering the reviewers comments and these edits, we propose the title: "*Dose Optimization of TBI-223 for Enhanced Therapeutic Benefit Compared to Linezolid in the BPaL Regimen.*” We are open to additional feedback.
2. The critique of the methods section is valid and aligns with feedback from Reviewer #1. We recognize the necessity for a clearer, more integrated description of our methods and have implemented extensive revisions to this section for enhanced clarity and thoroughness.

3. The experimental data were specifically assembled for this study to compare TBI-223 and LZD. However, the mouse and rabbit studies had separate objectives and were performed by separate laboratories (and hence additional harmonization of the methods was necessary). Further details are provided in the point-by-point responses.

General remark

- **Make sure to use either the term ‘non-clinical’ or ‘pre-clinical’ consistently throughout the manuscript.** We have changed all instances of nonclinical to preclinical

Abstract

- **“TBI-223 is a novel and more selective oxazolidinone...”. What is meant by the term ‘selective’. Is this the higher selectivity towards bacterial ribosomes instead of mitochondrial ribosomes? I suggest the authors either explain or remove this term.**

We have removed “selective”

- **“... and describe its efficacy and safety profile as compared to BPaL” – did you mean to compare TBI-223 to the BPaL regimen, or to compare TBI-223 to linezolid? In the last paragraph of the introduction the same sentence is used, but there it says “compared to linezolid”. Please clarify.**

The reviewer is correct, and we have changed this to “as compared to linezolid.”

- **The abstract is a nice summary of the work performed, but the direct aim of the study can be addressed more specifically.**

We have added “We aimed to optimize the dosing of TBI-223 within the BPaL regimen for enhanced therapeutic outcomes.” to the abstract and made minor changes to maintain word count.

- **Minor comment: the term “...nonclinical monotherapy” is not quite correct, as a single agent is never considered as treatment of active tuberculosis. It would be better to address this as “pre-clinical mono-exposure”.**

We have updated this text to preclinical

Introduction

- **“Linezolid dose reduction was necessary...” – it is interesting that the incidence of peripheral neuropathy and myelosuppression in the Nix-trial (81% and 48%, respectively) were much higher than in the ZeNix trial (38% and 22%, respectively) for the high dose 1200 mg 26 weeks arms. Are there any explanations known for this difference?**

We don’t have additional data to explain this. Speculatively, these were separate clinical trials. Nix-TB was a smaller study that only recruited 34 participants from 3 South African sites while Zenix enrolled 248 participants from 4 countries. The Zenix trial did occur after Nix-TB and may have had improved protocol interventions or better prepared trial sites. It’s plausible that these factors contributed to the different outcomes, but without specific comparative data, we cannot provide a definitive explanation for this variation.

- **“Using pharmacokinetics and safety data ... the therapeutic window of linezolid is quite narrow” – The point being made here is that linezolid has a narrow therapeutic window. However, the way this is explained is quite confusing:**

We have added short point by point corrections to the subpoints below, however, to address the overarching review of this text we’ve amended this paragraph as follows:

“Linezolid has a narrow therapeutic window, as evidenced by the concentration required for effective Mycobacterium tuberculosis eradication and the levels associated with patient toxicity. A pharmacokinetic-toxicodynamic model, that utilized population pharmacokinetic modeling to simulate dynamic individual

linezolid concentrations linked to toxicity models, identified a peripheral neuropathy IC₅₀ of 1.3 mg/L for linezolid (5). Median plasma concentrations from patients in the Nix-TB trial exceeded this threshold across all dosages, from 300 mg twice daily to 1200 mg once daily, with a minimum concentration of 2.1 mg/L at the lowest dose (5). These findings based on clinical data and modeling and simulations are consistent with previous studies that showed mitochondrial toxicity and safety events were associated with trough levels above 2 mg/L (6). Furthermore, the minimum inhibitory concentration (MIC) of linezolid necessary to inhibit 90% of Mycobacterium tuberculosis isolates range from 0.25 mg/L to 1.0 mg/L (7–10), underscoring the limited range within which linezolid can be both effective and safe.”

○ **Additional explanation is needed for the IC₅₀ of 1.3 mg/L. Where was this value obtained? A reference would be in place here.**

Reference 5 was added to this statement: 5. M. Z. Imperial, J. R. Nedelman, F. Conradie, R. M. Savic, Proposed Linezolid Dosing Strategies to Minimize Adverse Events for Treatment of Extensively Drug-Resistant Tuberculosis. *Clin. Infect. Dis. Off. Publ. Infect. Dis. Soc. Am.* **74**, 1736–1747 (2022).

○ **“This is consistent with previous studies that showed mitochondrial toxicity and safety events were associated with trough levels and susceptibility breakpoints greater than 2 mg/L”**

-- I do not understand how toxicity events can be associated with susceptibility breakpoints. I could not find a statement about this association in reference (7). That reference does state that the AUC₀₋₂₄ is linked to toxicity, but they also mention this could be because AUC₀₋₂₄ is collinear with trough levels. Please clarify.

This was based on reference 6 and we’ve removed reference 7. We agree that susceptibility breakpoint is incorrect and have removed this term as well.

○ **“Considering that the minimum inhibitory concentration of linezolid to cover 50% and 90% of Mycobacterium tuberculosis clinical isolates is 0.5 mg/L and 1.0 mg/L, respectively (8)” – this is quite a firm statement, based on only one study. However, when I look in reference (8), I am not sure where these values were obtained. 0.5 and 1.0 mg/L seem to be the MIC₅₀ and MIC₉₀ for Nocardia brasiliensis. The paper mentions an MIC₅₀ and MIC₉₀ for linezolid of 1 and 2 mg/L, respectively, but refers to another paper (PMID 16189119). Were these values also used as model parameters (see Table 1: MIC 0.5-1.0 mg/L)? Please check whether the values are correct.**

This reference is indeed incorrect, and we’ve amended the text to “minimum inhibitory concentration of linezolid to cover 90% of Mycobacterium tuberculosis clinical isolates ranges from 0.25 mg/L to 1.0 mg/L [7 - 10]”

7. Diacon AH, De Jager VR, Dawson R, et al. Fourteen-day bactericidal activity, safety, and pharmacokinetics of linezolid in adults with drug-sensitive pulmonary tuberculosis. *Antimicrob Agents Chemother* 2020; 64:e02012–19. Tato M, de la Pedrosa EG, Cantón R, et al. In vitro activity of linezolid against Mycobacterium tuberculosis complex, including multidrug-resistant Mycobacterium bovis isolates. *Int J Antimicrob Agents* 2006; 28:75–8.
8. Yang C, Lei H, Wang D, et al. In vitro activity of linezolid against clinical isolates of Mycobacterium tuberculosis, including multidrug-resistant and extensively drug-resistant strains from Beijing, China. *Jpn J Infect Dis* 2012; 65:240–2.

9. Zong Z, Jing W, Shi J, et al. Comparison of in vitro activity and MIC distributions between the novel oxazolidinone delpazolid and linezolid against multidrugresistant and extensively drug-resistant Mycobacterium tuberculosis in China. *Antimicrob Agents Chemother* 2018; 62:e00165–18.
10. Lopez B, Siqueira de Oliveira R, Pinhata JMW, et al. Bedaquiline and linezolid MIC distributions and epidemiological cut-off values for Mycobacterium tuberculosis in the Latin American region. *J Antimicrob Chemother* 2019; 74:373–9.

● **“We used a data-driven approach ... to provide safer, more effective TB regimens” – was safety indeed an outcome parameter? It is not discussed as such in the results section.**

We removed “safer”. While we believe that based on in vitro, in vivo and healthy volunteer study data that TBI-223 may be safer than linezolid, the reviewer is correct that it was not an outcome in our modeling.

● **“Additionally, we believe new and existing drugs regimens should be systematically evaluated for lesion distribution as achieving adequate exposures at the site of infection is crucial for successful treatment” -To be the devil’s advocate here – if treatment efficacy is already a primary outcome parameter, would it really be necessary to dive into the lesion distribution? I can imagine that when a certain drug regimen is less active than another it could be interesting to see whether this could be explained by a difference in tissue penetration, but what is the exact additive value of looking into tissue penetration in this study?** The reviewer's point is well-taken. Repeating the response for Reviewer #1 (1.388) on this topic; in this study, evaluating lesion distribution was crucial not to predict TBI-223's efficacy per se, but to discern if its tissue penetration could present an advantage or disadvantage compared to linezolid in a clinical scenario. Finding similar lesion penetration for both compounds streamlined our analysis, and ratios close to one suggested similar exposure levels in plasma and lesion which further allowed us to simplify our modeling approach. The similarities suggests that TBI-223's performance should be on par with linezolid, neither exceeding nor falling short in actual clinical settings. Additionally, our team is engaged in building comprehensive PK-PD models to determine inter- and intra-lesion efficacy, an extensive endeavor that falls beyond the scope of this current study. However, with similar exposures expected across lesions for both linezolid and TBI-223, it allowed us to reduce our assumptions to an apples-to-apples comparison using mouse PK-PD modeling to predict clinical outcomes.

We added the following to the discussion:

“Due to the similar penetration parameters and a penetration close to one for both study drugs, the final efficacy model was reduced to have only plasma concentrations drive efficacy. While the comparable lesion penetration of both compounds simplified our analysis, this similarity further implies that the clinical performance of TBI-223 should align closely with linezolid.”

Methods

● **Although I understand the authors have to comply to the word count restriction, quite a lot of useful information on the methodology of especially the animal experiments is missing. The authors refer to many other papers for the methods, but it would be good to give a brief summary so that the readers get a general idea of the methods used, instead of having to dive in to literature repeatedly. The LC/MS-MS methods on the other hand are described in much detail. Consider using the supplementary information for more in depth explanation on how the different studies were conducted. My comments below are in line with this general comment.**

Mouse infection and drug administration.

The following

• Useful information to add here would be (and this also accounts for the PK studies and rabbit studies):

○ How were the doses of TBI-223 and linezolid chosen?

For the mice studies doses to match linezolid clinical exposures and past experiments to refine dose response were used for linezolid doses. TBI-223 doses were scaled by 3 based on a 3-fold difference in vitro potency results. The methods text has been expanded to reflect this.

○ How big were the group sizes?

Approximately five mice per experimental group and the control group was equal to three. The rabbit experiments used 8 rabbits in total. These clarifications have been added to the manuscript.

○ What route of infection was used in the acute infection model?

Text expanded to include “high-dose aerosol infection” for the acute model.

○ How much time was there in between infection and start of treatment?

Added “incubation period of 6 days” to the text

○ How were the CFU-counts determined? On middlebrook agar (7H10 or 7H11?) with OADC? Were the samples washed or were charcoal-containing plates used to prevent drug carry-over?

Middlebrook 7H11 agar with 10% OADC was used. The following was added to the methods section:

“Mice were sacrificed 3 days after the last drug dose to reduce the risk of drug carryover. Lungs were collected and homogenized in glass grinders at pre-specified time points during and after drug treatment. The homogenates were serially diluted in PBS and plated on Middlebrook 7H11 agar plates supplemented with 10% (v/v) OADC (GIBCO) and cycloheximide [10 mg/mL], carbenicillin [50 mg/mL], polymixin B [25 mg/mL] and trimethoprim [20 mg/mL], except that media used in the acute infection model studies did not contain selective antibiotics.

Homogenates from mice treated with drug combinations were plated on the same agar media but with the addition of activated charcoal powder (0.4% w/v) to further prevent drug carryover. Colonies were counted after 4 and 6 weeks of incubation at 37°C to ensure all cultivable bacteria would be detected.”

○ Were humane endpoints considered?

Yes. Death was not an endpoint in our experiments. Animals were monitored at least twice daily for signs that would prompt early euthanasia. These included, but were not limited to, general signs of distress such as hunched posture, lethargy, anorexia, dehydration, rough hair coat and piloerection. Any mouse developing a moribund appearance (inactivity, hunched posture, ruffled fur, visible weight loss) or manifesting unusual behavior was euthanized by intentional isoflurane overdose followed by cervical dislocation. All procedures involving mice were approved by the Institutional Animal Care and Use Committee at Johns Hopkins University School of Medicine.

• Was the MIC of TBI-223 and linezolid determined for the H37Rv? Or was the range of MIC (as indicted in Table 1) used?

A range of MIC’s were considered that included clinical isolates from literature and in house data using H37Rv.

• Were the pharmacodynamic-studies with BPaL really necessary to conduct, or could ‘historical data’ also have been used here?

The PD studies were essential for obtaining directly comparable data with TBI-223. Although PK-PD modeling could account for experimental disparities, establishing this translational platform required minimizing procedural variability to the greatest extent possible at these early stages of implementation. Our group is currently working on multiple studies to look at historical data and normalizing experimental conditions via modeling to address this concern in future work. Some examples of this ongoing work are below:

1. Jacqueline P. Ernest, Janice Jia Ni Goh, Natasha Strydom, Qianwen Wang, Rob C. van Wijk, Nan Zhang, Amelia Deitchman, Eric Nuernberger, Rada M. Savic. *European Respiratory Journal* 2023; DOI: 10.1183/13993003.00165-2023

2. Qianwen Wang, Janice JN Goh, Nan Zhang, Eric Nuermberger, Rada Savic. Prospectively predicting BPamZ Phase IIb outcomes using a translational preclinical mouse to human platform. bioRxiv 2023.02.16.528876

• **The authors mention ‘PK-PD data’, but were pharmacokinetics assessed in these experiments? It seems only PD was studied in these studies, as there is no information on blood sampling, time points, or plasma concentration determinations.**

The methods section has been restructured and a section on “Pharmacokinetics experiments” has been added. *“Pharmacokinetic experiments*

Comprehensive toxicology studies in mice, rats and dogs were performed for TBI-223 and linezolid as part of the initial IND filing. Pharmacokinetic data were collected during the experiments which were used for a multispecies pharmacokinetic model that predicted human projected concentrations for the Phase 1 trial of TBI-223. Appendix A describes this multispecies pharmacokinetic model in more detail. This pharmacokinetic model was repurposed to predict mouse pharmacokinetics for the PD studies described below.

Pre-clinical PK data were obtained from uninfected mice, rats and beagle dogs in experiments performed at BioDuro, Inc. For the animal procedures, male C57BL/6 and BALB/c mice were purchased from Lingchang biotechnology Co., Ltd (Shanghai, China), Sprague-Dawley rats from Beijing Vital River laboratories and beagle dogs from Beijing Marshall Biotechnology.,Ltd. All animals were housed in cages with food and water available optionally. Laboratory conditions included a clean environment at 20 to 25°C under 50–60% humidity and 12 h/12 h light/dark cycles. All the procedures involving animals were approved by the Institutional Animal Care and Use Committee (IACUC).

Fed female BALB/c mice weighing approximately 25 g received single linezolid or TBI-223 doses of 5 mg/kg by tail vein injection (IV) or 100, 250 or 500 mg/kg by oral administration (gavage). C57BL/6 mice weighing approximately 30 g received TBI-223 5 mg/kg by tail vein injection or 100 mg/kg by gavage. Sprague-Dawley rats weighing approximately 250 g received linezolid 3 mg/kg by tail vein injection or 100 mg/kg by gavage. Beagle dogs weighing approximately 10 kg received linezolid or TBI-223 doses of 3 mg/kg by tail vein injection (IV) or 25 mg/kg by gavage or TBI-223 100, 250 or 500 mg/kg by gavage. Plasma was sampled at 5 minutes (IV only), 0.25, 0.5, 1, 2, 4, 7 and 24 hr post-dose. Linezolid and TBI-223 were quantified by high pressure liquid chromatography coupled to tandem mass spectrometry (LC/MS-MS). Protein precipitation extraction (PPT) was performed by adding acetonitrile (ACN) containing internal standard (IS, terfenadine) to 1 volume of plasma. The PPT mixtures were vortexed for 1 min and centrifuged at 4,000 rpm for 15 min. The supernatant was then transferred for LC/MS-MS analysis.

The LC/MS-MS analysis was performed on a Sciex Applied Biosystems API 4000 triple-quadrupole mass spectrometer coupled to a Shimadzu LC-20AD to quantify the samples. Chromatography was conducted using an Agilent Kinetex 2.6 µm C18 100A column (3 × 50 mm) under a reverse-phase gradient elution. The aqueous mobile phase (A) comprised 0.1% formic acid in Milli-Q deionized water and the organic mobile phase (B) contained 0.1% formic acid in acetonitrile. The gradient was initiated with 8:2 for A to B at a flow rate of 0.8 mL/min, maintained until 0.30 min. From 0.30 to 1.50 min, the percentage of A was decreased to 10%. This composition was held constant until 2.00 min. Immediately after, the gradient was reverted back to 8:2 for one minute to re-equilibrate the column. Multiple-reaction monitoring (MRM) of parent/daughter transitions in electrospray positive-ionization mode was used to quantify the analytes. The following MRM transitions were used respectively for linezolid (338.20/296.00), TBI-223 (366.27/296.10) and terfenadine (472.40/436.40). Sample analysis was accepted if the concentrations of the quality control samples were within 20% of the nominal concentration. Data processing was performed using Analyst software (version 1.5.1; Applied Biosystems Sciex).”

• **Quite a lot of PK experiments were performed with various animals (mice, rats, beagle dogs). Were all these experiments really necessary for the aim of this study? Were all these data used as input for the**

predictive model? For example, why was PK after intravenous administration tested, as both linezolid and TBI-223 can be dosed orally in animals and humans?

The extensive PK experiments, while not essential for this specific study, were integral to the GLP toxicology studies for TBI-223's IND submission. In the interest of not repeating mouse PK experiments or additional modeling, the multispecies TBI-223 model used to inform the human PK predictions for the TBI-223 Phase 1 trial was used to predict mouse drug levels in the PD experiments. We've added this background to the manuscript.

• In line with this, considering the 3Rs of Reduction, Refinement, and Replacement, did the aim of the study truly justify the numbers of animals used and discomfort levels in these studies? Or was this experiment secondary to a primary study within which separate efficacy was assessed and described?

These animal studies were primarily conducted prior to and in alignment with the TBI-223's IND submission, focusing on a direct toxicity comparison with linezolid. Consistent with the 3Rs principle, no additional PK data were collected during the PD studies, minimizing animal use and discomfort.

Rabbit lesion experiments

• Antimycobacterial and lesion-specific concentration targets

○ Please provide extra information on the rationale and methods used for these experiments to guide the readers.

Methods now include “Rabbits were selected for the lesion experiments for their consistency in developing lung cavities that mirror the complex pathology of human pulmonary cavitary tuberculosis.” Additional text has been added as suggested point by point below.

○ Why are these studies placed within the ‘rabbit lesion experiments’-section?

The studies fall under the 'rabbit lesion experiments' section for two primary reasons: first, their results directly influenced the determination of coverage in the lesion experiments, integrating them into that result reporting. Second, the experiments utilized caseum derived specifically from the rabbit models. We have restructured this section to clarify these were within the rabbit experiment framework and conducted with collected tissue from the rabbit experiments.

• Rabbit infection and drug administration

○ Why is information provided on animal weight and housing conditions here, but not for the mouse drug activity studies and for the PK studies in mice, rats and beagles?

We have added the same information to the other sections.

○ How was the human-equivalent efficacious dose calculated? ‘Based on modeling’ is quite blunt.

Added the following to this methods section:

The following text was added “To provide the human-equivalent dose in rabbits, plasma PK models were built using the initial PK data from these experiments and doses calculated that would provide similar rabbit drug exposures compared to human exposures.”

○ How was the human-equivalent efficacious dose for TBI-223 determined? Based on the clinical phase I data?

TBI-223 used the same approach as LZD and the text has been amended to:

“The human-equivalent efficacious dose was estimated to be 90 mg/kg for linezolid (equivalent to 1200 mg once daily) and 200 mg/kg for TBI-223 (equivalent to 2400 mg once daily)”

○ The human equivalent doses were determined in uninfected rabbits, while the tissue penetration experiments were executed in infected animals. Does infection influence the PK of linezolid and TBI-223?

The PK of drugs could certainly be influenced by infection. We didn't see an impact in these particular data for TBI-223 and LZD.

○ **How many rabbits were used in these experiments?**

The text has been updated to clarify that eight rabbits were used per experiment.

○ **Why were rabbits chosen for these studies? Could using C3HeB/FeJ (Kramnik) mice also have been an option as they develop necrotizing granulomas as well?**

Rabbits were selected for these studies due to their consistent development of lung cavities that closely mimic the pathological features observed in human pulmonary cavitary TB. The Kramnik mouse model is a valuable model due to the formation of necrotizing granulomas, but we have seen better consistency in lesion development with rabbit studies. We have added the following statement to the methods:

“Rabbits were selected for the lesion experiments for their consistency in developing lung cavities that mirror the complex pathology of human pulmonary cavitary tuberculosis (22).”

22. D. Rifat, B. Prideaux, R. M. Savic, M. E. Urbanowski, T. L. Parsons, B. Luna, M. A. Marzinke, A. A. Ordonez, V. P. DeMarco, S. K. Jain, V. Dartois, W. R. Bishai, K. E. Dooley, Pharmacokinetics of rifapentine and rifampin in a rabbit model of tuberculosis and correlation with clinical trial data. *Sci. Transl. Med.* **10**, eaai7786 (2018).

○ **Was drug activity not assessed in the rabbit studies?**

Drug activity was not directly assessed in these specific rabbit studies. Our focus was primarily on understanding the distribution of the drugs. However, we are actively conducting separate longitudinal efficacy studies in rabbits to evaluate drug activity for multiple next generation TB drugs. This translational platform is currently ongoing and constitute a separate body of work.

General modelling workflow

● **Although figure 1 depicts a comprehensive overview of the modelling approach that was used, the general set-up of the study is still quite difficult to understand. Was the model based on already performed animal experiments? Or were the animal experiments chosen such that clinical outcome can be optimally predicted? And in this case, how is decided what experiments need to be performed in order to be able to make predictions about TBI-223's clinical activity? A paragraph explaining this would better guide the readers through the study.**

The following was added to the Discussion section:

“The research strategy to provide this integrative approach was structured around a systematic selection of animal experiments, each chosen for its ability to yield data critical for predicting clinical outcomes. This included understanding TBI-223's efficacy in monotherapy and combination therapies and its pharmacodynamics at the infection site. The model was therefore an integration of data from these key animal studies, forming a composite framework that utilized in vivo data for clinical activity prediction and early clinical data, Figure 1.”

● **“After data collation, a nonclinical PK model...” – typo in ‘collation’; also, please clarify ‘response data’.**

We believe there may be a misunderstanding as the term 'collation' is used here in the context of gathered and organizing data.

We have expanded the text to: “PK model was first established to link time-dependent concentration to response data that included lesion site of action concentrations and separately bacterial load in the mice efficacy models.”

● **Two assumptions in the model are mentioned in the first paragraph (on the PK-PD response in mice versus humans and on the ratio of penetration parameters) – can this be substantiated with references?**

We have added the following references:

22. D. Rifat, B. Prideaux, R. M. Savic, M. E. Urbanowski, T. L. Parsons, B. Luna, M. A. Marzinke, A. A. Ordonez, V. P. DeMarco, S. K. Jain, V. Dartois, W. R. Bishai, K. E. Dooley, Pharmacokinetics of rifapentine and rifampin in a rabbit model of tuberculosis and correlation with clinical trial data. *Sci. Transl. Med.* **10**, eaai7786 (2018).

27. N. Zhang, N. Strydom, S. Tyagi, H. Soni, R. Tasneen, E. L. Nuermberger, R. M. Savic, Mechanistic modeling of Mycobacterium tuberculosis infection in murine models for drug and vaccine efficacy studies. *Antimicrob. Agents Chemother.* **64** (2020), doi:10.1128/AAC.01727-19.

• **“For culture conversion simulations culture conversion was assumed to be reached at the moment the bacterial compartment reached 1 bacterium” – On what information is this assumption based? Would that mean that a patient’s sputum culture would turn positive when there are 2 bacteria present in the lungs? This might be quite a strict assumption, as it does not match the technical sensitivity/lower limit of detection of mycobacterial cultures in practice.**

The reviewer is correct that this is a strict assumption and does not match the technical sensitivity/LOD of Mtb in practice. However it is an assumption that seems to work in this context and has translated well in previous models from our group (PMID: 32810158). We would also clarify that it’s not necessarily 2 bacteria in the lung, but 2 bacteria in the sample taken from a patient. Originally this was selected on the mathematical basis that less than one bacterium is not reasonably possible. Speculatively perhaps 1 bacterium in a sample and the probability of sampling 1 bacterium is enough for regrowth in the culture conversion test. The model is ultimately based on real world data, and while we cannot detect below the limit of detection, the model appears to account for the conditions that predict culture conversion.

• **“Clinical, rabbit, mouse plasma PK estimation and model building included...” – does this mean that the PK information obtained in rats and beagles was not used in the modelling?**

They were used for the PK model that informed mouse PK-PD. The expanded text in the methods hopefully clarifies this.

• **“For monotherapy EBA simulations efficacy parameters ... intrinsic resistance expression or multiple other factors” – were chronic mouse model experiments only performed for linezolid, or also for TBI-223?**

For the chronic mouse model experiments, only linezolid data from historical experiments were utilized. In our combination modeling, we explored previous data from acute, subacute, and chronic models for linezolid to assess its potential for refining combination estimates. However, a model without priors effectively predicted BPaL results.

The methods text has been updated as follows:

“Additionally methods that estimated EC_{50} with fixed E_{max} parameters from monotherapy experiments, including acute, subacute and chronic infection models were attempted as a way to estimate an EC_{50} shift or correction factor for oxazolidinone when bedaquiline and pretomanid were added. However this constraint did not provide reasonable fits and EC_{50} and E_{max} were estimated independently without monotherapy priors.”

• **“For cellular lesions the macrophage IC_{90} value was used as the most appropriate target and caseous lesions used caseum MBC_{90} .” – can you explain why these parameters were most appropriate best for each type of lesion?**

We’ve expanded this text as follows:

“Mycobacterium tuberculosis populations residing in different tissue microenvironments are phenotypically distinct and respond differently to drug treatment and therefore antimicrobial targets that replicate these microenvironments were used. Specifically for cellular lesions the macrophage IC_{90} values were used as the most appropriate target and caseous lesions used caseum MBC_{90} .”

Results

• **In the first paragraph the authors refer to Appendix A – I think this should be Supplementary figure 1?**

Apologies, Appendix A was accidentally not added in the initial submission. It is included in the revised version

● **“The experiments used an acute 6-day infection model ... to 1000 mg/kg for TBI-223” – this information seems more in place in the method section.**

We have corrected that these specifics are related to describing figure 2

● **Subhead 2: comparison of linezolid predictions with clinical endpoints**

○ **The authors state twice that the model ‘performed well’. Can you explain when you would say a model performs ‘well’? As a reader with limited background in modelling it is difficult to estimate whether this can indeed be considered as a ‘well-performing’-model.**

We appreciate the reviewer's point for clarity. In the revised manuscript, we specify that a "well-performing" model as evidenced by our Visual Predictive Check (VPC), where 90% prediction intervals accurately captured the observed data's 5th, 50th, and 95th percentiles, indicating reliable and unbiased predictions

Specific text changes:

“The model performed well at predicting bactericidal activity as determined by CFU measurements in combination with bedaquiline and pretomanid in the first 28 days of treatment in the Nix-TB trial, Fig. 3B, with majority of the observed data falling within the 95% prediction intervals, and the median of the observed data aligning closely with the prediction median of the model”

“In Fig. 3C, simulations of sputum culture conversion are depicted alongside observed Nix-TB trial data in a Kaplan-Meier curve. The model accurately reflected the progression of culture conversion, matching the observed probabilities of positive status with those predicted at individual time points.”

○ **“observations were within range of the simulations” (Figure 3E) – am I correct that the range of the simulations is rather large?**

Correct, the simulation range appears large due to substantial individual variability in LZD pharmacokinetics, evident in the plasma panel. Additionally, the small trial size contributed to this variability, highlighted by a participant with concentration levels 10-fold higher than others in the lesion panels. These real-world data and variability are reflected in the simulations, which incorporate interindividual variability estimates derived from actual patient data.

● **Subhead 4: dose recommendations for TBI-223 and its comparison to linezolid**

○ **“To evaluate the treatment duration-shortening potential of TBI-223 in combination” – this is a tricky statement, as the authors did not look at the sterilizing activity of TBI-223 (or linezolid) alone or in combination. It is questionable if short term outcome parameters such as culture conversion can be used as a proxy for treatment shortening potential.**

We've updated this sentence to “To evaluate the impact of TBI-223 in combination with bedaquiline and pretomanid on culture conversion”

Figure 1:

- **“collect dose ranging preclinical data from suitable animal models” – what makes an animal model suitable?**

“Suitable” was removed from the text but in general animal models that have consistency of responses to drug interventions and/or are able to replicate human disease pathology to some degree.

- **In the ‘Clinical prediction’-panel – what is the difference between the red and the blue lines?** Corrected in caption: “linezolid (red) and TBI-223 (blue).”

- **Can you explain the ‘Lesion distribution and coverage’-figures? What are we looking at here?**

Additional information is provided in Figure 8 where final results are shown:

Lesion coverage is defined as lesion-specific drug concentrations above the respective lesion target concentration for each hour, Figure S8. Each colored square represents 1 h that a drug is above its respective PD target. Empty blocks show drug concentration below the relevant monotherapy PD target and blue and navy

blocks represent hours above target concentration for TBI-223 2400 mg daily and 1200 mg twice daily respectively and red blocks for linezolid (LZD). The selected monotherapy PD targets for TBI-223 were MIC for uninvolved lung (2 mg/L), macrophage IC₉₀ for cellular lesions (4.2 mg/L) and caseum MBC₉₀ for caseum (46.8 mg/L).

Figure 2:

- The caption is very brief. Considering the broad public of Nature Communications, the authors might want to elaborate on these figures. I imagine a tremendous amount of work was put in all the animal studies, and these deserved to be well-understood.

We have expanded the captions to provide additional information to help with the dense figures

Figure 4:

- Please explain the abbreviations: IR, SR, IC₉₀.

Abbreviations were added to the caption

Supporting information:

- Consider adding figure captions for S3-S6. These are difficult to interpret without explanation.

Captions have been added for these figures

Discussion

• **The authors have written a well-reasoned discussion that places their findings in a relevant broader context. It is very nice to read that this study omitted the need for phase 2A monotherapy studies.** We thank the reviewer for their kind comments.

• **In the paragraph starting with “A 2400 mg dose of TBI-223 is predicted to produce concentrations...” the safety of TBI-223 is addressed, which is a highly relevant topic, since toxicity of linezolid is the main issue of the BPaL regimen as discussed elaborately in the introduction. However, it is not clear whether safety of TBI-223 was assessed within this study. Table 1 shows ‘safety margin’ as one of the model parameters, but was safety assessed in the experiments performed in the current study, or was this parameter based on a previously executed toxicity study in rats? The results section focusses on anti-tuberculosis activity of TBI-223 and lesion penetration, but toxicity of TBI-223 is not discussed. To understand whether the proposed dose of 2400 mg TBI-223 would actually be feasible in terms of toxicity, it would be good to elaborate on this topic.**

The toxicology of TBI-223 was initially evaluated during the IND submission and shared as an abstract at the ASM Microbe conference in 2017 (1). We used these data to calculate safety margins based on the multi-species PK model and the clinical population PK model (developed during our Phase 1 study). Importantly, we did not perform any additional toxicology studies in animals. Recognizing the reviewer's concerns about the ambiguity surrounding the reporting of toxicity and PK study sources, we've amended the methods section to better specify the history behind this work.

Furthermore, concerning safety discussions, we acknowledge the term “safety” is most often reserved for clinical data. To ensure clarity and present a balanced view of TBI-223's potential, we've tempered language around “safety” (also highlighted by the editor).

1. K. Mdluli, C. Cooper, T. Yang, M. Lotlikar, F. Betoudji, M. Pinn, P. Converse, E. Nuermberger, S.-N. Cho, T. Oh, X. Liu, D. Zhang, H. Huang, N. Fotouhi, TBI-223: A Safer Oxazolidinone in Pre-Clinical Development for Tuberculosis (available at <https://www.abstractsonline.com/pp8/#!/4358/presentation/6174>).

- **“we predict that TBI-223 will provide more favorable long-term safety and efficacy outcomes” -- ‘Long-term efficacy’ implies that the study looked at the ability of TBI-223 to cure TB without relapse. However, this study only addresses ‘short-term’ clinical outcomes such as EBA, time to culture conversion and time to positivity. The authors might want to rephrase this sentence and mention that ‘long-term’ outcome parameters such as cure without relapse of infection were not studied here. This can be seen as a limitation of the study, since short-term clinical outcomes such as EBA do not necessarily predict relapse-free curation.**

We have updated this sentence as follows: “If no dose interruptions are necessary, we predict that TBI-223 may provide more favorable efficacy outcomes when compared to linezolid in combination with bedaquiline and pretomanid”

REVIEWERS' COMMENTS

Reviewer #2 (Remarks to the Author):

The authors invested substantial efforts into improving the manuscript and putting together a comprehensive response to all comments made, which is much appreciated. The previously raised comments are adequately and thoroughly addressed, especially refining the safety statement, providing a more detailed elaboration on the methodology, and clarification of the origin and relevance of the various PK data, and lesion PK data in particular. The adjustments significantly improved the overall clarity of this important study. I look forward to seeing the final version of the manuscript published.